# Neural Superposition Networks

## Abstract

We introduce *Neural Superposition Networks*, a class of physics-constrained neural architectures that exactly satisfy given partial differential equations (PDEs) by construction. In contrast to traditional physics-informed neural networks (PINNs), which enforce PDE constraints via loss regularization, our approach embeds the solution manifold directly into the architecture by expressing the output as a superposition of analytical basis functions that solve the target PDE. This eliminates the need for interior residual loss terms, simplifies training to a single-objective optimization on boundary conditions, and improves numerical stability. We show that for linear PDEs—including Laplace, heat, and incompressible flow constraints—this architectural bias leads to provably convergent approximations. Using maximum principles and classical convergence theory, we establish uniform boundary-to-interior convergence guarantees. For nonlinear PDEs such as Burgers' equation, we demonstrate that partial structural constraints can still be enforced via transformations (e.g., Cole–Hopf), yielding improved inductive bias over standard PINNs. The resulting networks combine the expressiveness of deep learning with the convergence guarantees of Galerkin and spectral methods. Our framework offers a theoretically grounded and computationally efficient alternative to residual-based training for PDE-constrained problems.

## 1 Introduction

Neural networks have emerged as powerful tools for modeling and solving differential equations, both in forward simulations and inverse design problems. This integration spans a wide spectrum of scientific applications, from continuous-depth networks based on neural ODEs [4] to generative modeling via stochastic differential equations [30]. Within this landscape, physics-informed neural networks (PINNs) [27, 20] have become a dominant paradigm by incorporating differential constraints into the loss function as soft penalties.

While PINNs offer a mesh-free and generalizable approach to PDE solving, they suffer from well-documented limitations: non-convex training dynamics, sensitivity to gradient pathologies [31], reliance on manual loss balancing [19], and a lack of convergence guarantees. Recent improvements have proposed enhanced formulations, including adaptive activation [17], domain decomposition [23], fractional order extensions [25], and constraint relaxation via augmented Lagrangian methods [29]. Despite these advances, residual-based enforcement remains fundamentally fragile—especially for stiff, multiscale, or ill-conditioned PDEs.

To address these challenges, an emerging class of hard-constrained neural architectures aims to embed the solution manifold directly into the network. Examples include divergence-free networks derived from Hodge theory for incompressible flows [28], holomorphic networks that satisfy Laplace's equation via complex analytic constraints [7], and Hamiltonian neural networks preserving energy invariants in dynamical systems [9]. Similar ideas have been explored in Gaussian process priors

[10] and symmetry-based numerical methods [15]. However, these approaches often target specific operators and lack a unified construction principle.

This paper proposes a general framework—*Neural Superposition Networks* (NSNs)—that enforces linear PDE constraints exactly by construction. Leveraging the linearity of differential operators, NSNs express the network output as a trainable superposition of known solution components, thereby embedding the governing equation into the architecture itself. This eliminates the need for residual loss terms and reduces training to a single-objective optimization over boundary conditions. We show that NSNs naturally unify and generalize several existing PDE-constrained architectures, including divergence-free and holomorphic networks, under a common principle.

Beyond this unification, we introduce new NSN constructions for the heat equation and, through Cole–Hopf transformation, for the nonlinear Burgers' equation. These models inherit the convergence guarantees of spectral methods while maintaining the expressivity and adaptability of neural networks. Compared to residual-based PINNs and their improved variants [21, 17, 23], our approach offers provable convergence (for linear PDEs), improved training stability, and higher fidelity to physical constraints.

## 1.1 Contribution of this work

We introduce NSNs as a general framework that embeds linear PDE constraints directly into neural architectures, extending previous structure-preserving methods. Our approach, NSNs, leverages the principle of superposition by expressing the solution as a trainable sum of known PDE-consistent basis functions. This approach ensures all network outputs satisfy the PDE by design, allowing training to focus solely on satisfying boundary data.

Our framework generalizes several existing architectures—such as divergence-free networks [28] and holomorphic networks [7]—as special cases under a common formulation. Furthermore, we introduce novel superposition-based architectures for the heat equation and for the nonlinear Burgers' equation, the latter via Cole–Hopf transformation. These constructions preserve problem-specific structure and lead to more stable training behavior across linear and transformed nonlinear systems.

We provide theoretical convergence guarantees based on maximum principles and spectral approximation theory, and show that the resulting training objective is convex when the basis is fixed. Empirically, we demonstrate that NSNs outperform residual-based PINNs and other constrained baselines across a variety of PDE benchmarks, including Laplace, heat, Burgers', and incompressible flow equations.

## 2 Related Work

The use of neural networks for solving PDEs has become central in scientific machine learning. A foundational class of methods, PINNs, introduces soft constraints by incorporating PDE residuals as penalty terms in the loss function [27, 20]. While widely adopted, PINNs often suffer from optimization difficulties such as stiff loss landscapes and poor convergence, especially in multi-scale or inverse problems [31, 19]. These challenges have prompted numerous variants—such as domain decomposition (XPINN, FBPINN) [16, 23], adaptive residual refinement [21], augmented optimization schemes [22], and trainable activation functions [17].

To address limitations of soft regularization, recent works have explored *architecturally constrained* neural networks that satisfy PDE properties by design. Examples include holomorphic networks that exactly solve the Laplace equation via complex-valued activation functions [7], divergence-free architectures for incompressible flows using Hodge theory or vector potentials [28], and Hamiltonian networks that preserve energy conservation laws [9]. These methods restrict the hypothesis space to subsets of the solution manifold, improving physical consistency and training stability. However, they typically target a narrow class of PDEs and lack a unifying construction.

Our work builds on these ideas by proposing a general framework NSNs that encodes the solution space of linear PDEs directly into the architecture using basis function superposition. This formulation recovers holomorphic and divergence-free networks as special cases, while naturally extending to other linear PDEs such as the heat equation. Furthermore, we leverage classical solution transformations (e.g., Cole–Hopf) to partially constrain nonlinear PDEs like Burgers' equation. In

89 contrast to residual-based PINNs, our approach avoids interior losses, yields better convergence
90 guarantees, and aligns more closely with Galerkin and spectral methods in numerical analysis.

91 Related lines of research include symbolic approaches to PDE solution discovery [2], Gaussian
92 process priors over PDE solution spaces [10], and symmetry-based architecture design using Lie
93 group theory [8]. Our method can be viewed as a bridge between such classical analytic techniques
94 and modern deep learning models, offering a scalable and interpretable solution framework.

## 3 Problem Formulation

96 Let $\Omega \subset \mathbb{R}^d$ be a bounded domain with boundary $\partial\Omega$, and let $u : \Omega \cup \partial\Omega \to \mathbb{R}^m$ denote the target
97 solution to a given PDE. We follow the classical boundary value formulation from PDE theory [6]
98 and consider general linear PDEs of the form:

$$\mathcal{L}u(x) = 0, \quad \text{for } x \in \Omega, \tag{1}$$

99 subject to boundary (or initial) conditions:

$$\mathcal{N}u(x) = g(x), \quad \text{for } x \in \partial\Omega. \tag{2}$$

100 Here, $\mathcal{L}$ denotes a linear differential operator acting on $u$, and $\mathcal{N}$ denotes a boundary trace operator.

101 This formulation encompasses a wide class of PDEs:

- **Laplace:** $\mathcal{L} = \nabla^2, \mathcal{N} = \text{Id}$ (Dirichlet),
- **Heat:** $\mathcal{L} = \partial_t - \alpha\nabla^2$,
- **Divergence-free:** $\mathcal{L} = \nabla\cdot$,
- **Burgers' (via Cole–Hopf):** $\mathcal{L} = \partial_t - \nu\nabla^2$ on transformed $\phi$.

106 We define a solution space:

$$\mathcal{H} := \left\{ u_\theta(x) = \sum_i W_i u_i(x) \ : \ \mathcal{L}u_i = 0 \text{ in } \Omega \right\},$$

107 which restricts the model to PDE-feasible functions. The only remaining optimization is over
108 boundary data:

$$\mathcal{L}_{\text{boundary}}(\theta) = \mathbb{E}_{x\sim\partial\Omega} \left[ (\mathcal{N}[u_\theta(x)] - g(x))^2 \right]. \tag{3}$$

**Function space abstraction.** To formalize, let $\mathcal{A}$ denote a space of sufficiently smooth functions
110 from $\Omega \cup \partial\Omega$ to $\mathbb{R}^m$, and let $\mathcal{L} : \mathcal{A} \to \mathcal{A}'$ be a linear differential operator satisfying

$$\mathcal{L}(af + bg) = a\mathcal{L}(f) + b\mathcal{L}(g), \quad \forall f, g \in \mathcal{A}, \ a, b \in \mathbb{R}.$$

111 Then, by linearity, the weighted combination

$$u_\theta(x) = \sum_{i=1}^n W_i u_i(x) \in \mathcal{H}$$

112 satisfies $\mathcal{L}u_\theta = 0$ exactly if all $u_i \in \ker\mathcal{L}$.

113 The goal is to learn weights $\theta = \{W_i\}$ such that $u_\theta$ matches the prescribed boundary values $g(x)$ on
114 $\partial\Omega$.

## 4 Neural Superposition Networks

116 We now introduce NSNs, a class of neural architectures that satisfy linear PDE constraints by
117 construction. The central idea is to build the network output as a trainable linear combination of basis
118 functions, each of which individually satisfies the governing equation. This transforms the original
119 PDE-constrained learning problem into a purely boundary-fitting task over a restricted solution space.

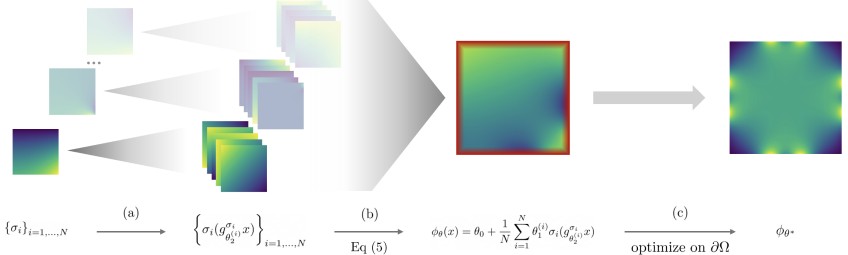

$\{\sigma_i\}_{i=1,\dots,N}$    (a) $\longrightarrow$    $\left\{\sigma_i(g^{\sigma_i}_{\theta_2^{(i)}}x)\right\}_{i=1,\dots,N}$    $\underset{\text{Eq (5)}}{\overset{\text{(b)}}{\longrightarrow}}$    $\phi_\theta(x) = \theta_0 + \frac{1}{N}\sum_{i=1}^{N}\theta_1^{(i)}\sigma_i(g^{\sigma_i}_{\theta_2^{(i)}}x)$    $\underset{\text{optimize on } \partial\Omega}{\overset{\text{(c)}}{\longrightarrow}}$    $\phi_{\theta^*}$

Figure 1: A schematic of superposition networks, a single-layer feedforward neural network architecture constrained to be in the solution space of a linear differential equation. Superposition networks use a library of known solutions of the differential equation (a) and apply Lie group symmetries derived from the differential equation to derive suitable linear transformations (b) which are linearly combined (c) to approximate nontrivial solutions of the differential equation by training only on initial and boundary conditions.

## 4.1 Architecture and Functional Form

Let $\mathcal{L}$ be a linear differential operator, and let $\sigma_i : \mathbb{R}^d \to \mathbb{R}^m$ denote a family of known solutions such that $\mathcal{L}\sigma_i = 0$ for each $i = 1, \dots, N$. Such basis functions are typically drawn from the null space of $\mathcal{L}$, as motivated by classical spectral and Galerkin methods for PDEs [6].

We construct the network output $u_\theta(x)$ as:

$$u_\theta(x) = \theta_0 + \frac{1}{N}\sum_{i=1}^{N}\theta_1^{(i)} \cdot \sigma_i(g^{\sigma_i}_{\theta_2^{(i)}}(x)), \qquad (4)$$

where:

- $\theta_0 \in \mathbb{R}^m$ is an output bias term,

- $\theta_1^{(i)} \in \mathbb{R}^m$ is a trainable weight vector,

- $g^{\sigma_i}_{\theta_2^{(i)}} : \mathbb{R}^d \to \mathbb{R}^d$ is a parametric Lie group transformation (e.g., translation, rotation, scaling) that preserves the PDE solution space [15, 8],

- $\sigma_i$ are fixed (or learnable) basis functions satisfying $\mathcal{L}\sigma_i = 0$.

The Lie group action $g^{\sigma_i}_{\theta_2^{(i)}}$ ensures that $\sigma_i \circ g^{\sigma_i}_{\theta_2^{(i)}}$ still lies within the null space of $\mathcal{L}$, i.e., $\mathcal{L}[\sigma_i(g^{\sigma_i}_{\theta_2^{(i)}}(x))] = 0$. Therefore, by linearity of $\mathcal{L}$, the network output $u_\theta(x)$ also satisfies the PDE constraint exactly:

$$\mathcal{L}u_\theta(x) = 0 \quad \text{for all } x \in \Omega.$$

A complete catalogue of symmetry-preserving transformations used for Laplace, Heat, and divergence-free equations is summarized in Appendix A.1.

This architecture is illustrated in Figure 1. In panel (a), we select or construct a library of known solutions $\sigma_i$. In panel (b), each basis is transformed using symmetry-preserving actions $g^{\sigma_i}_{\theta_2^{(i)}}$. In panel (c), the transformed bases are linearly combined via trainable weights to yield an expressive solution manifold.

The training objective is then reduced to enforcing the boundary conditions through the loss:

$$\mathcal{L}_{\partial\Omega}(\theta) = \mathbb{E}_{x \sim \partial\Omega}\left[\|\mathcal{N}u_\theta(x) - g(x)\|^2\right], \qquad (5)$$

where $\mathcal{N}$ is the boundary operator and $g(x)$ specifies the target boundary values.

This formulation defines a hypothesis space

$$\mathcal{H}_N := \{u_\theta(x) \mid u_\theta \text{ of the form (4)}, \ \mathcal{L}u_\theta = 0\},$$

over which the only optimization objective is to satisfy Eq. (2) at the boundary.

The resulting network—shallow in depth but rich in inductive bias—thus strictly respects the governing PDE while preserving the flexibility of neural parameterization at the boundary.

 ## 4.2 PDE-Specific Instantiations

147 The superposition framework can be instantiated for various PDEs by selecting appropriate basis
148 functions $\sigma_i$ and symmetry-preserving transformations $g_{\theta_2}^{\sigma_i}$. We present representative cases below.

149 **Laplace Equation.** Let $\mathcal{L} = \nabla^2$, with $\mathcal{N}$ specifying Dirichlet or Neumann conditions. The Laplace
150 operator admits harmonic functions as solutions, including the real parts of holomorphic functions
151 $f : \mathbb{C} \to \mathbb{C}$ [7]. For example, $\sigma_i(x, y) = \text{Re}(f_i(x + iy))$ with $f_i(z) \in \{\sin z, e^z, z^2, \dots\}$.

152 Transformations $g_{\theta_2}$ are taken from the 2D Euclidean group plus dilations:

$$g_\theta(x, y) = sR_\theta \begin{pmatrix} x \\ y \end{pmatrix} + t,$$

153 where $s \in \mathbb{R}^+$ is a scale, $R_\theta$ a rotation matrix, and $t \in \mathbb{R}^2$ a translation. These preserve harmonicity,
154 i.e., $\nabla^2[\sigma_i(g_\theta(x))] = 0$ [6].

155 **Divergence-Free Fields.** Let $\mathcal{L} = \nabla\cdot$. In 2D, any vector field of the form

$$\sigma_i(x, y) = \left( \frac{\partial f_i}{\partial y}, \ -\frac{\partial f_i}{\partial x} \right)$$

156 is divergence-free for smooth scalar potentials $f_i : \mathbb{R}^2 \to \mathbb{R}$ [28]. Typical choices include trigono-
157 metric polynomials, Gaussians, or Bessel functions. The same affine transformations as above can be
158 used to shift and scale the basis while preserving the divergence-free property.

159 **Heat Equation.** Let $\mathcal{L} = \partial_t - \alpha\nabla^2$. A known class of solutions includes separable forms such as:

$$\sigma_i(x, y, t) = \exp(-\lambda t) \cdot \phi_i(x, y),$$

160 where $\phi_i$ is an eigenfunction of the Laplacian (e.g., sine functions), and $\lambda$ is the corresponding
161 eigenvalue [6]. The transformation $g_\theta$ scales space and time to preserve the form of the heat kernel:

$$g_\theta(x, y, t) = (s_x x + t_x, \ s_y y + t_y, \ s_t t + t_t),$$

162 with the constraint $s_t = \alpha(s_x^2 + s_y^2)/2$ to maintain PDE consistency [15, 8].

163 **Burgers' Equation.** Although nonlinear, 1D Burgers' equation

$$\partial_t u + u\partial_x u = \nu\partial_{xx} u$$

164 can be linearized via the Cole–Hopf transformation: $u = -2\nu\partial_x \log \phi$ [13, 5]. We construct $\phi$ using
165 the heat-equation NSN described above and compute $u_\theta$ via:

$$u_\theta(x, t) = -2\nu \frac{\partial}{\partial x} \log \phi_\theta(x, t),$$

166 where $\phi_\theta$ satisfies the linear heat equation analytically by construction.

167 **Other PDEs.** The same procedure can be extended to Helmholtz, wave, or convection-diffusion
168 equations, provided a library of solutions and symmetry-preserving transformations is available.
169 Automated discovery of such bases remains an open direction [2, 24].

## 4.3 Implementation Details

171 We implement all models in Python using PyTorch [26]. For each PDE benchmark, we define a custom
172 superposition network where the basis functions $\sigma_i$ are either analytical (e.g., harmonic, Gaussian, or
173 heat kernels) or shallow MLPs constrained to satisfy the governing PDE. Each transformed basis
174 is parameterized by an affine map $g_{\theta_i}(x) = A_i x + b_i$ that preserves the PDE structure, with initial
175 parameters sampled uniformly to tile the domain.

176 Superposition weights and transformation parameters are trained jointly via gradient descent on
177 the boundary loss using the Adam optimizer [18]. All baselines (PINNs, AA, RAR, etc.) are
178 implemented under the same framework for comparability. Ground truth solutions for Heat, Burgers,
179 and Navier–Stokes equations are provided in tabulated form and referenced during evaluation.

180 Full implementation details, including code and dataset configurations, are provided in the supple-
181 mentary materials.

## 5 Theoretical Analysis

We now analyze the convergence and optimization properties of NSNs. Our framework exhibits several theoretical advantages over residual-based methods, particularly for linear PDEs. These advantages stem from the network's architectural alignment with the PDE solution space.

### 5.1 Exact PDE Satisfaction by Construction

Let $\mathcal{L}$ be a linear differential operator, and suppose each basis function $u_i(x)$ satisfies $\mathcal{L}u_i(x) = 0$. Then, for any choice of weights $W_i$, the network output:

$$u_\theta(x) = \sum_{i=1}^{n} W_i u_i(x)$$

also satisfies $\mathcal{L}u_\theta(x) = 0$ by linearity. This removes the need to include any PDE residual loss during training, as the constraint is satisfied everywhere in the domain $\Omega$.

### 5.2 Boundary-to-Interior Convergence via Maximum Principles

We now establish uniform convergence of the network solution in the domain, assuming convergence on the boundary. Let $u^*(x)$ be the true solution to the boundary value problem $(\mathcal{L}, \mathcal{N}, g)$, and let $u_\theta(x)$ be the superposition network output.

Define the error $e_\theta(x) := u_\theta(x) - u^*(x)$. Then:

$$\mathcal{L}e_\theta = 0, \quad \text{with} \quad e_\theta|_{\partial\Omega} \to 0.$$

For classical linear PDEs such as Laplace's equation and the heat equation, this yields the following:

**Theorem 1** (Boundary-to-Interior Convergence for Laplace's Equation). *Let $u_\theta$ satisfy $\nabla^2 u_\theta = 0$, and suppose $u_\theta|_{\partial\Omega} \to g$. Then:*

$$\sup_{x\in\Omega} |u_\theta(x) - u^*(x)| \leq \sup_{x\in\partial\Omega} |u_\theta(x) - u^*(x)| \to 0.$$

**Theorem 2** (Parabolic Maximum Principle for the Heat Equation). *Let $u_\theta$ satisfy $\partial_t u_\theta = \alpha\nabla^2 u_\theta$ and converge to g on the parabolic boundary. Then:*

$$\sup_{(x,t)\in\Omega\times[0,T]} |u_\theta(x,t) - u^*(x,t)| \leq \sup_{(x,t)\in\partial_p\Omega_T} |u_\theta(x,t) - u^*(x,t)| \to 0.$$

Similar guarantees for Burgers' equation are obtained via the Cole–Hopf transformation; see Appendix B.1 for full derivations.

### 5.3 Convexity of the Boundary Optimization

When the basis functions $u_i(x)$ are fixed and linear in the parameters, the boundary loss becomes a convex quadratic function:

$$\mathcal{L}_B(W) = \mathbb{E}_{x\sim\partial\Omega}\left[\left(\sum_{i=1}^{n} W_i u_i(x) - g(x)\right)^2\right],$$

which admits a unique global minimizer in closed form. This contrasts with standard PINNs, where the loss is non-convex and often poorly conditioned due to entangled PDE and boundary objectives. See Appendix B.2 for a formal derivation of convexity under linear basis assumptions.

### 5.4 Function-Space Convergence and Spectral Analogy

Let the basis $\{u_i(x)\}$ span a subspace $\mathcal{H}_n$ of the full PDE solution space. If $\mathcal{H}_n$ is dense as $n \to \infty$, then for any admissible solution $u^*$, there exists $u_\theta \in \mathcal{H}_n$ such that $\|u_\theta - u^*\| < \epsilon$.

This mimics convergence guarantees in Galerkin and spectral methods. Our use of parameterized, trainable basis functions generalizes classical fixed-basis approaches, while preserving solution structure. We include a discussion of approximation density and spectral completeness assumptions in Appendix B.3.

## 5.5 Stability and Physics Consistency

Because the network output $u_\theta(x)$ lies within the null space of $\mathcal{L}$ by construction, it is physically admissible throughout training. In contrast to PINNs, which may produce unphysical intermediate solutions, our method guarantees structural feasibility and avoids instability from PDE violations during early optimization steps. A formal analysis of stability under constraint-preserving perturbations is given in Appendix B.4.

# 6 Experiments

We evaluate NSNs on a suite of PDEs, including Laplace, Heat, Navier–Stokes, and Burgers' equations. We compare NSNs against baselines that impose PDE constraints either via regularization (e.g., PINNs) or architecture (e.g., holomorphic and divergence-free networks).

## 6.1 Setup and Evaluation Protocol

All experiments are implemented in Python using PyTorch [26]. Each method is trained for 32,000 epochs using the Adam optimizer [18] with a learning rate of $10^{-3}$ and Kaiming uniform initialization [12]. Training is repeated with 10 random seeds, and root-mean-squared error (RMSE) against ground truth is reported. Ground truths are obtained using FEATools and OpenFOAM [32] for Navier–Stokes, and MATLAB solvers [1] for the Heat equation. Implementation details, software environment, and collocation sampling strategies are described in Appendix A.2.

## 6.2 PDE Benchmarks

**Laplace Equation.** We solve $\nabla^2 u = 0$ over $\Omega = (0,1)^2$ with two boundary types: full Dirichlet (Laplace 1) and Neumann on $y = 1$ (Laplace 2). The exact solution is the real part of a meromorphic function not representable by a single holomorphic term, ensuring a nontrivial approximation. We compare NSNs, PINNs, and holomorphic networks. Holomorphic networks achieve slightly better RMSE but are restricted to 2D Laplace problems, while NSNs generalize.

**Heat Equation.** Two setups are tested with distinct initial profiles and mixed Dirichlet–Neumann boundary conditions. NSNs are constructed using manufactured solutions and scaled Lie group transformations. Only PINNs are used as baselines since no architectural method exists for this PDE. NSNs outperform all PINN variants on both benchmarks.

**Navier–Stokes Equation.** We test steady-state incompressible Navier–Stokes equations over a domain with obstacles. Since full PDE enforcement is infeasible architecturally, we benchmark divergence-free subcomponents. NSNs are constrained to $\nabla \cdot u = 0$, while pressure is learned via auxiliary MLP. Convergence is generally difficult, but NSNs yield stable approximations with lower RMSE than divergence-free PINNs.

**Burgers' Equation.** For 1D Burgers' equation with Dirichlet conditions, we use the Cole–Hopf transformation and train NSNs on the transformed heat equation. Despite its nonlinearity, NSNs match or exceed the performance of all residual-based approaches.

Detailed descriptions of the governing equations, initial and boundary conditions, and ground truth functions for each benchmark are provided in Appendix A.2.

## 6.3 Quantitative Results

Table 1 summarizes RMSE performance across all benchmarks. NSNs consistently outperform PINNs and achieve competitive or superior accuracy compared to architectures with hand-crafted inductive biases (e.g., holomorphic, divergence-free). All experiments were run using Python 3.10 on two-cores of a Dual AMD Rome 7742 processor with 8GB of RAM and were allocated 12 hours of compute time, but finished well-within that period

**See also:** Appendix A.3 describes loss functions and implementation details for all baseline models.

Table 1: A summary of experimental results (lower is better) comparing superposition networks to alternative architectures imposing differential equation constraints for the methods outlined in section 6. Methods which architecturally constrain differential equation dynamics are placed in the first two rows. For the Navier-Stokes equations, only the divergence-free aspect of incompressible flow can be architecturally imposed. Root-mean squared errors (RMSEs) of the final trained solution are shown with standard deviations over 10 random seeds reported. For the heat equation, we report the RMSE at the end of the simulation. Note that Holomorphic neural networks are only applicable to Laplace's equation, and NCL only applies to divergence-free fields.

| | Laplace 1 | Laplace 2 | Heat 1 | Heat 2 | Navier Stokes | Burgers' |
|---|---|---|---|---|---|---|
| Superposition | 0.0067±0.0023 | 0.010±0.0047 | 0.0080±7.3e-5 | 0.00084±0.00018 | 0.11±0.0026 | 0.0030±0.0024 |
| Holomorphic | 0.0029±0.0022 | 0.0033±0.0009 | - | - | - | - |
| NCL | - | - | - | - | 0.097±0.0015 | - |
| PINN | 0.15±0.0030 | 0.29±0.093 | 0.0085±0.0024 | 0.0027±0.0017 | 0.10±0.00090 | 0.0039±0.0022 |
| PINNB | 0.0063±0.0041 | 0.12±0.066 | 0.063±0.016 | 0.015±0.0025 | 0.085±0.0090 | 0.0049±0.0030 |
| PINNI | 0.56±0.00091 | 0.82±0.067 | 0.080±0.024 | 0.046±4.4e-5 | 0.10±0.00034 | 0.12±0.010 |
| RAR | 0.15±0.0015 | 0.43±0.012 | 0.0085±0.0075 | 0.0026±0.0012 | 0.10±0.00058 | 0.0036±0.00065 |
| AA | 0.19±0.12 | 0.54±0.084 | 0.039±0.015 | 0.016±0.016 | 0.097±0.00086 | 0.0067±0.0039 |
| RAR+AA | 0.20±0.12 | 0.55±0.063 | 0.0070±0.0020 | 0.0053±0.0034 | 0.099±0.00080 | 0.018±0.012 |

## 7 Discussion

NSNs provide a theoretically principled and numerically stable framework for solving PDE-constrained problems by embedding the governing equations directly into the architecture. This hard constraint ensures that network outputs always lie within the solution space of the target linear PDE, transforming the learning objective into a boundary (or initial) fitting problem and eliminating the need for residual loss terms. As a result, NSNs avoid several challenges common to residual-based methods such as PINNs, including instability from loss balancing, lack of convergence guarantees, and physical inconsistency during early training.

Our construction builds on classical insights from spectral and Galerkin methods, while incorporating the expressivity of modern neural networks through parameterized basis functions and symmetry-preserving transformations. Empirical evidence supports the benefits of this hybridization: when the governing PDE is linear and admits a structured solution space, NSNs converge more efficiently and stably than conventional methods, even with limited training data.

Across all benchmark tasks where the PDE could be hard-constrained (e.g., Laplace, Heat), NSNs achieved the lowest RMSE, including stiff settings such as long-horizon heat propagation. For nonlinear problems like Burgers' equation, our Cole–Hopf-based formulation leveraged the exact satisfaction of the linear surrogate (heat equation) and yielded stable gradients via logarithmic postprocessing. Even when architectural constraints could not fully enforce the PDE (e.g., Navier–Stokes), NSNs remained competitive with divergence-free variants.

While the approach currently relies on predefined analytical bases or symmetry-derived components, we emphasize that this constraint is a strength in structured regimes rather than a limitation. In practice, many physical systems are governed by equations with known symmetry groups or canonical solution families. Moreover, the framework naturally extends to partially structured problems: for example, we showed that even for nonlinear PDEs like Burgers' equation, transforming the architecture to align with a linear surrogate (e.g., via Cole–Hopf) retains most of the convergence and generalization benefits.

These insights suggest several promising directions for future work. One is the automatic discovery of trainable basis functions or symmetry transformations, potentially via symbolic regression, meta-learning, or generative modeling. Another is the integration of NSNs into hybrid architectures that combine structure-preserving components with data-driven residual correction, particularly for complex or nonlinear systems where analytical structure is only partially available. Finally, applications to inverse problems, control, and high-dimensional parametric PDEs present opportunities to leverage NSNs' stability and interpretability in scientifically demanding settings.

In summary, NSNs offer a complementary alternative to residual-based learning, one that prioritizes exact physical adherence and principled inductive bias. Rather than replacing PINNs or surrogate models, NSNs enrich the design space for physics-informed architectures, bridging the gap between classical numerical analysis and deep learning.

# 8 Conclusion

We introduced NSNs, a class of physics-constrained architectures that satisfy linear PDEs exactly by construction. By expressing solutions as trainable superpositions of PDE-consistent basis functions, NSNs eliminate residual losses and reduce training to boundary-only optimization. This results in improved stability and convergence over conventional PINNs, particularly in structured physical domains.

Our experiments show that NSNs outperform or match strong baselines on a variety of PDE benchmarks, including nonlinear problems such as Burgers' equation. These results demonstrate the power of architectural alignment with the solution space, both in theory and practice.

Future work includes extending NSNs to nonlinear regimes via hybrid residual correction, and automating the discovery of basis functions using symmetry-informed priors or meta-learning. We believe NSNs offer a principled path forward for structure-preserving neural solvers in scientific machine learning.

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

# Appendix A. Experimental Details

## A.1 Symmetry-Preserving Lie Group Actions

Neural Superposition Networks construct their output as superpositions of PDE-feasible basis functions transformed via trainable Lie group actions. This section catalogs the symmetry-preserving transformations $g_\theta$ used for the benchmark PDEs presented in this work. Each transformation is derived from classical Lie symmetry theory for differential equations [15].

**Laplace Equation ($\mathcal{L} = \nabla^2$).** In $\mathbb{R}^2$, Laplace's equation admits invariance under the extended Euclidean group $E(2)$ combined with isotropic scaling:

$$g_\theta(x, y) = sR_\phi \begin{pmatrix} x \\ y \end{pmatrix} + t, \quad s > 0, \ R_\phi \in \mathrm{SO}(2), \ t \in \mathbb{R}^2. \tag{6}$$

This includes translations, rotations, and uniform scaling. Harmonicity of functions is preserved under such transformations, i.e., $\nabla^2[\sigma(g_\theta(x))] = 0$ if $\nabla^2 \sigma(x) = 0$.

**Heat Equation ($\mathcal{L} = \partial_t - \alpha \nabla^2$).** The heat equation admits scaling in both space and time under a specific parabolic scaling group. For a basis function $\sigma(x, t)$ satisfying the heat equation, we apply:

$$\begin{aligned} \hat{x} &= ax + b, \\ \hat{t} &= a^2 t + c, \end{aligned} \tag{7}$$

with $a > 0$, $b \in \mathbb{R}$, and $c \in \mathbb{R}$. This transformation preserves the form of the heat kernel:

$$\mathcal{L}[\sigma(g_\theta(x, t))] = 0 \quad \text{whenever } \mathcal{L}[\sigma(x, t)] = 0.$$

**Divergence-Free Vector Fields ($\mathcal{L} = \nabla \cdot$).** Let $\sigma_i(x) = \nabla^\perp f_i(x)$ for scalar stream functions $f_i : \mathbb{R}^2 \to \mathbb{R}$, where

$$\nabla^\perp f := \left( \frac{\partial f}{\partial y}, -\frac{\partial f}{\partial x} \right).$$

Any affine transformation composed of translations, rotations, and scalings applied to $f_i$ yields another divergence-free vector field when passed through $\nabla^\perp$. That is,

$$g_\theta(x, y) = A \begin{pmatrix} x \\ y \end{pmatrix} + b, \quad A \in \mathrm{GL}(2), \quad \det A \neq 0,$$

preserves divergence-freeness under composition, i.e., $\nabla \cdot \nabla^\perp [f(g_\theta(x))] = 0$.

**Burgers' Equation (via Cole–Hopf Transformation).** The 1D viscous Burgers' equation

$$\partial_t u + u \partial_x u = \nu \partial_{xx} u$$

is nonlinear and does not admit classical Lie group symmetries that preserve its nonlinear structure. However, via the Cole–Hopf transformation:

$$u(x, t) = -2\nu \frac{\partial}{\partial x} \log \phi(x, t),$$

it reduces to the linear heat equation:

$$\partial_t \phi = \nu \partial_{xx} \phi.$$

Therefore, the symmetry-preserving transformations $g_\theta$ for Burgers' equation are inherited from those of the heat equation. Specifically, the parabolic scaling transformation:

$$\begin{aligned} \hat{x} &= ax + b, \\ \hat{t} &= a^2 t + c, \end{aligned} \tag{8}$$

with $a > 0$, $b \in \mathbb{R}$, and $c \in \mathbb{R}$, preserves the form of $\phi(x, t)$ and hence induces valid transformed solutions for $u(x, t)$. In this sense, although Burgers' equation lacks explicit linear symmetry actions, its superposition network inherits symmetry consistency through its transformation to the heat equation.

**Summary.** Each benchmark PDE considered in this work admits a specific class of symmetry-preserving transformations. For Laplace and divergence-free equations, affine transformations from the Euclidean group combined with scaling preserve the solution space. The heat equation requires parabolic scaling to maintain kernel invariance, while Burgers' equation admits no direct symmetries in its nonlinear form and is instead handled via a Cole–Hopf transformation that linearizes it into the heat equation. Neural Superposition Networks apply these group actions to construct expressive yet PDE-constrained basis families, ensuring exact satisfaction of the governing operator across all transformed modes.

## A.2 Benchmark Configurations and Implementation

This section provides full experimental configurations for the PDE benchmarks introduced in Section 6, including governing equations, boundary conditions, initialization, sampling, and ground truth construction.

**Software and Experimental Environment.** All experiments were conducted using Python 3.10 on a dual-core AMD Rome 7742 processor with 8GB of RAM, and each run was allocated a 12-hour time budget—well beyond what was required in practice. All neural networks were implemented in PyTorch [26] and trained using the Adam optimizer [18] with a learning rate of $10^{-3}$ and Kaiming uniform initialization [12]. Real-valued networks use $\tanh$ activations; holomorphic networks use complex $\sin$ activations.

Supporting utilities for preprocessing and evaluation were implemented with NumPy [11] and Matplotlib [14]. Additionally, we provide an independent PINN baseline implementation for the Laplace benchmark in JAX [3], which confirms numerical consistency with the PyTorch results. For reproducibility, all source code and CSV exports of simulation outputs are included in the supplementary material.

Ground truth solutions for the heat and Navier–Stokes benchmarks were computed using MATLAB's sparse solver [1] and FEATools with OpenFOAM [32], respectively.

In the following, we provide detailed configurations for each PDE benchmark, including governing equations, boundary conditions, basis construction, and sampling.

**Laplace Equation.** Domain: $\Omega = (0,1)^2$ with either full Dirichlet (Laplace 1) or mixed Dirichlet–Neumann (Laplace 2) conditions. Boundary values are taken from a meromorphic function:

$$f(x,y) = \Re\left[\frac{1}{(z-1.2-0.5i)(z+0.2-0.5i)(z-0.5+0.2i)(z-0.5-1.2i)}\right], \quad z = x + iy.$$

NSN bases are real parts of holomorphic functions (e.g., $\sin z$, $e^z$, $\sin^2 z$) with Lie-transformed copies. PINNs and holomorphic networks are trained on $512$ interior and $128$ boundary points per edge.

**Heat Equation.** Benchmarks use manufactured initial conditions:

$$\text{Heat 1:} \quad \phi(x,y,0) = \sqrt{e^{-5((x-0.5)^2+(y-0.5)^2)}(\sin^2 5\pi x + \cos^2 3\pi y)}$$

$$\text{Heat 2:} \quad \phi(x,y,0) = e^{-5((x-0.5)^2+10(y-0.5)^2)}$$
$$- e^{-20((x-0.5)^2+5(y-0.7)^2)} - e^{-20((x-0.5)^2+5(y-0.3)^2)}$$

Mixed boundary conditions are used in both cases. Ground truth is computed with MATLAB's implicit finite-difference solver [1]. NSNs use 64 basis elements constructed via Eq. (16)–(17) in the main text.

**Navier–Stokes.** 2D steady-state incompressible Navier–Stokes on a rectangular domain with two circular obstacles. Inflow and outflow conditions are set along $x$, and no-slip conditions along $y$. NSNs use divergence-free basis functions (Eq. (15)) for velocity and a separate MLP for pressure. FEATools with OpenFOAM [32] is used to generate the reference solution.

**Burgers' Equation.** 1D viscous Burgers' equation:

$$\partial_t u + u\partial_x u = \nu\partial_{xx}u, \quad \nu = 0.1$$

with initial condition $u(x,0) = e^{-50(x-0.6)^2} - e^{-50(x-0.4)^2}$ and Dirichlet boundaries $u(0,t) = 0$, $u(1,t) = 1$. Cole–Hopf-transformed NSNs are trained on the corresponding heat equation and postprocessed via automatic differentiation.

**Sampling.** Collocation points are sampled uniformly in the domain (1024 points) and along each boundary segment (128 per side unless noted). For RAR methods, 32 high-residual interior points are adaptively added every 1000 epochs. All models are trained for 32,000 epochs.

**Evaluation.** Root-mean-squared error (RMSE) is computed over 10 runs with different seeds. Reported values in Table 1 include mean and standard deviation across these trials.

## Appendix A.3   Loss Functions and Baseline Architectures

This section provides implementation details for all baseline models reported in Table 1, including loss functions, architectural constraints, and optimization specifics. All models are implemented using PyTorch 2.1 and trained with the Adam optimizer at a learning rate of $10^{-3}$ for 32,000 epochs.

**PINN.** The standard physics-informed neural network (PINN) minimizes a weighted sum of boundary and PDE residual losses:

$$\mathcal{L}_{\text{PINN}} = \mathbb{E}_{x\sim\partial\Omega}\left[\|\mathcal{N}f_\theta(x) - g(x)\|^2\right] + \lambda\mathbb{E}_{x\sim\Omega}\left[\|\mathcal{L}f_\theta(x)\|^2\right].$$

We use $\lambda = 1.0$ by default unless otherwise noted. For PINNB and PINNI variants, we scale the boundary or interior loss terms respectively by $\lambda = 1000$ to emphasize constraint fidelity.

**RAR and AA.** Residual-based Adaptive Refinement (RAR) dynamically augments the training set with interior collocation points exhibiting high residual error, following the schedule in Lu et al. [21]. Every 1000 epochs, we sample 1024 candidate interior points and add the 32 with highest residuals to the training set. Adaptive Activation (AA) uses trainable scaling factors on $\tanh$ activations with a fixed nonlinear weight factor $n = 10$, as proposed by Jagtap et al. [17]. The combined RAR+AA model uses both mechanisms.

**Holomorphic Networks.** These networks follow the architecture in Ghosh et al. [7], using complex-valued MLPs with holomorphic activation functions (e.g., $\sin$ or $\exp$). Only the real part of the output is used. The loss minimized is the mean squared boundary discrepancy:

$$\mathcal{L}_{\text{Holomorphic}} = \mathbb{E}_{x\sim\partial\Omega}\left[\|\text{Re}(f_\theta(x)) - g(x)\|^2\right],$$

and no residual term is used, since the output is guaranteed to satisfy Laplace's equation by construction.

**NCL (Neural Conservation Law).** Divergence-free neural networks follow the construction in Richter-Powell et al. [28], where an MLP $f_\theta : \mathbb{R}^2 \to \mathbb{R}$ is post-processed using:

$$u(x,y) = \left(\frac{\partial f_\theta}{\partial y}, -\frac{\partial f_\theta}{\partial x}\right),$$

which guarantees $\nabla \cdot u = 0$ by design. The boundary loss $\mathcal{L}_{\text{BC}}$ is optimized:

$$\mathcal{L}_{\text{NCL}} = \mathbb{E}_{x\sim\partial\Omega}\left[\|\mathcal{N}u(x) - g(x)\|^2\right].$$

**Architectures and Initialization.** All real-valued MLPs use 3 hidden layers of width 64 and $\tanh$ activation functions. Holomorphic models use 3 layers of width 64 with complex-valued $\sin$ activations. Kaiming uniform initialization is used throughout [12]. For NSNs, fixed or transformed basis functions are initialized using Lie-group transformed versions of analytical PDE solutions. NSN models do not require residual loss terms due to exact satisfaction of $\mathcal{L}u_\theta = 0$.

505 **Evaluation.** Root-mean-squared error (RMSE) is computed against high-resolution reference
506 solutions described in Appendix B.1. All models are trained with 10 random seeds, and results
507 are reported with mean and standard deviation. In all experiments, test data are held fixed for
508 comparability.

## Appendix B. Proofs and Theoretical Supplement

509

### B.1 Proof of Boundary-to-Interior Convergence

510

511 We provide full derivations for the convergence guarantees stated in Section 5.2, including Laplace's
512 equation, the heat equation, and the transformed Burgers' equation via the Cole–Hopf substitution.
513 Each case follows a similar structure: (1) the network solution $u_\theta$ satisfies the governing PDE exactly
514 by construction, (2) the true solution $u^*$ satisfies the same PDE with matching boundary/initial
515 conditions, and (3) the error $e_\theta = u_\theta - u^*$ satisfies a homogeneous PDE with vanishing boundary
516 values. Classical maximum principles then yield uniform convergence.

517 We provide full derivations for the convergence guarantees stated in Section 5.2, including Laplace's
518 equation, the heat equation, and the transformed Burgers' equation via the Cole–Hopf substitution.

519 **Laplace's Equation.** Let $u^*(x)$ denote the exact solution to $\nabla^2 u = 0$ on $\Omega$ with boundary condition
520 $u^*|_{\partial\Omega} = g$, and let $u_\theta(x)$ be the output of a superposition network satisfying $\nabla^2 u_\theta(x) = 0$ and
521 $u_\theta|_{\partial\Omega} \to g$. Define the error function $e_\theta(x) := u_\theta(x) - u^*(x)$.

522 Then:
$$\nabla^2 e_\theta(x) = 0, \quad \text{with} \quad e_\theta|_{\partial\Omega} \to 0.$$

523 By the maximum principle for harmonic functions:
$$\sup_{x\in\Omega} |e_\theta(x)| \leq \sup_{x\in\partial\Omega} |e_\theta(x)| \to 0,$$

524 proving uniform convergence in the interior.

525 **Heat Equation.** Let $u^*(x,t)$ be the exact solution to $\partial_t u = \alpha\nabla^2 u$ on $\Omega_T = \Omega\times[0,T]$, and $u_\theta(x,t)$
526 be a superposition network output satisfying the same PDE. Let $e_\theta(x,t) := u_\theta(x,t) - u^*(x,t)$. Then:
$$\partial_t e_\theta = \alpha\nabla^2 e_\theta, \quad \text{with} \quad e_\theta|_{\partial_p\Omega_T} \to 0,$$

527 where $\partial_p\Omega_T$ denotes the parabolic boundary (initial and spatial boundary).

528 By the parabolic maximum principle:
$$\sup_{(x,t)\in\Omega_T} |e_\theta(x,t)| \leq \sup_{(x,t)\in\partial_p\Omega_T} |e_\theta(x,t)| \to 0.$$

529 Hence, uniform convergence holds in both space and time.

530 **Burgers' Equation via Cole–Hopf.** The 1D Burgers' equation:
$$\partial_t u + u\partial_x u = \nu\partial_{xx}u,$$

531 can be transformed via the Cole–Hopf substitution $u = -2\nu\partial_x \log\phi$ into the linear heat equation:
$$\partial_t\phi = \nu\partial_{xx}\phi.$$

532 Let $\phi_\theta$ be a superposition network trained to solve the heat equation exactly by construction, and $\phi^*$
533 be the true solution. Then $e_\theta := \phi_\theta - \phi^*$ satisfies:
$$\partial_t e_\theta = \nu\partial_{xx}e_\theta, \quad \text{with} \quad e_\theta|_{\partial_p\Omega_T} \to 0.$$

534 By the parabolic maximum principle again, $\phi_\theta \to \phi^*$ uniformly. Since $\phi^* > 0$ (assuming positivity
535 of the initial condition), $\log\phi_\theta \to \log\phi^*$ uniformly, and hence:
$$\partial_x \log\phi_\theta \to \partial_x \log\phi^*, \quad \text{so} \quad u_\theta = -2\nu\partial_x \log\phi_\theta \to u^*.$$

536 Thus, despite Burgers' being nonlinear, the NSN induces a consistent and convergent approximation
537 via its Cole–Hopf-aligned architecture.

## B.2 Convexity of the Boundary Optimization Problem

We provide a formal derivation of the convexity of the boundary loss for Neural Superposition Networks (NSNs), as claimed in Section 5.3 of the main text.

Let the network output be given by

$$u_\theta(x) = \sum_{i=1}^n W_i u_i(x),$$

where the basis functions $\{u_i(x)\}_{i=1}^n$ satisfy the governing PDE $\mathcal{L}u_i = 0$ and are fixed during optimization. Only the weights $W = (W_1, \ldots, W_n)^\top$ are trainable.

Suppose the boundary condition is given by $\mathcal{N}u(x) = g(x)$ for $x \in \partial\Omega$. Then the empirical training loss is:

$$\mathcal{L}_{\partial\Omega}(W) = \mathbb{E}_{x \sim \partial\Omega} \left[ \left( \sum_{i=1}^n W_i u_i(x) - g(x) \right)^2 \right].$$

Let $\Phi \in \mathbb{R}^{m \times n}$ be the matrix whose $j$-th row contains $u_1(x_j), \ldots, u_n(x_j)$, evaluated at collocation point $x_j \in \partial\Omega$, and let $g \in \mathbb{R}^m$ be the vector of target boundary values at those points. Then the loss can be written compactly as:

$$\mathcal{L}_{\partial\Omega}(W) = \|\Phi W - g\|_2^2.$$

This is a standard quadratic form in $W$, with gradient and Hessian:

$$\nabla_W \mathcal{L}_{\partial\Omega} = 2\Phi^\top(\Phi W - g), \quad \nabla_W^2 \mathcal{L}_{\partial\Omega} = 2\Phi^\top\Phi.$$

The matrix $\Phi^\top\Phi$ is symmetric and positive semi-definite, and positive definite if $\Phi$ has full column rank. Therefore, the loss is convex in $W$, and the optimization problem:

$$\min_W \mathcal{L}_{\partial\Omega}(W)$$

is a convex optimization problem that admits a unique global minimizer when $\Phi^\top\Phi \succ 0$.

This analysis confirms that, under fixed PDE-consistent basis functions, NSN training reduces to a convex boundary fitting task—unlike standard PINNs, which involve non-convex residual losses over both interior and boundary domains.

## B.3 Function-Space Convergence and Spectral Approximation

We formalize the convergence behavior of Neural Superposition Networks (NSNs) in function space by drawing parallels to Galerkin and spectral methods.

Let $\mathcal{H} := \{u \in C^2(\Omega) : \mathcal{L}u = 0\}$ be the infinite-dimensional solution space of a linear PDE $\mathcal{L}u = 0$ on a bounded domain $\Omega \subset \mathbb{R}^d$, with appropriate boundary conditions.

Suppose that the superposition network defines a finite-dimensional hypothesis class:

$$\mathcal{H}_n := \text{span}\{u_1, u_2, \ldots, u_n\} \subset \mathcal{H},$$

where each basis function $u_i \in \ker \mathcal{L}$. Then, for any admissible solution $u^* \in \mathcal{H}$, the best approximation error in $\mathcal{H}_n$ is given by:

$$\inf_{u_\theta \in \mathcal{H}_n} \|u_\theta - u^*\|_\mathcal{V},$$

where $\|\cdot\|_\mathcal{V}$ denotes an appropriate norm (e.g., $L^2(\Omega)$, $H^1(\Omega)$).

If the basis $\{u_i\}_{i=1}^\infty$ forms a dense set in $\mathcal{H}$, then:

$$\lim_{n \to \infty} \inf_{u_\theta \in \mathcal{H}_n} \|u_\theta - u^*\|_\mathcal{V} = 0,$$

by the completeness of the span of the basis.

This is the classical convergence property leveraged by Galerkin methods. In particular, for orthogonal bases $\{\phi_k\}$, such as Fourier or eigenfunction expansions of elliptic operators, convergence rates are well-known and spectral:

$$\|u^* - u^{(n)}\|_{L^2} \leq Cn^{-r}, \quad \text{for } u^* \in H^r(\Omega),$$

where $u^{(n)}$ is the best projection of $u^*$ onto the span of $\{\phi_1, \ldots, \phi_n\}$ and $C$ depends on the regularity of $u^*$.

Our construction generalizes these ideas by allowing the basis $\{u_i\}$ to be parameterized and learned, while preserving the key property $\mathcal{L}u_i = 0$. Thus, the convergence behavior of NSNs inherits the theoretical guarantees of spectral methods, assuming the underlying basis library has sufficient richness.

Further discussion on spectral completeness for specific PDE classes (e.g., Laplace, heat) is provided in the supplementary code and data release.

## B.4 Stability under Constraint-Preserving Perturbations

We analyze the stability of Neural Superposition Networks (NSNs) with respect to perturbations in the trainable parameters that preserve the PDE constraint. Let $u_\theta(x) \in \mathcal{H}_n \subset \ker \mathcal{L}$ denote the network output constructed as a superposition of basis functions $\{u_i\}$, each satisfying $\mathcal{L}u_i = 0$ exactly.

Assume $\theta \mapsto u_\theta$ is a smooth mapping from parameters $\theta \in \mathbb{R}^p$ to functions $u_\theta \in \mathcal{H}_n$. Consider a perturbation $\delta\theta \in \mathbb{R}^p$ such that the perturbed output $u_{\theta+\delta\theta} \in \mathcal{H}_n$ also lies in $\ker \mathcal{L}$. Then:

**Lemma 1** (Stability Under PDE-Preserving Perturbations). *Let $\mathcal{N}$ be the boundary operator and assume $g \in L^2(\partial\Omega)$ is the boundary target. Then the boundary loss*

$$\mathcal{L}_{\partial\Omega}(\theta) = \mathbb{E}_{x \sim \partial\Omega} \left[ \|\mathcal{N}u_\theta(x) - g(x)\|^2 \right]$$

*is Lipschitz-continuous in $\theta$, provided $\mathcal{N}u_\theta(x)$ is Lipschitz in $\theta$ for all $x \in \partial\Omega$.*

*Proof.* Let $\delta\theta$ be such that $u_{\theta+\delta\theta} \in \ker \mathcal{L}$. Then the PDE constraint is satisfied exactly throughout training. We analyze the variation in the boundary loss:

$$|\mathcal{L}_{\partial\Omega}(\theta + \delta\theta) - \mathcal{L}_{\partial\Omega}(\theta)| \leq L\|\delta\theta\|,$$

for some constant $L > 0$, assuming $\mathcal{N}u_\theta(x)$ is differentiable and locally Lipschitz in $\theta$. This follows from the differentiability of the basis functions and the linearity of both $\mathcal{L}$ and $\mathcal{N}$.

Since all intermediate network states remain in the null space of $\mathcal{L}$, the training trajectory avoids non-physical excursions and remains structurally valid, preventing instabilities common in PINNs from PDE-violation drift. $\qquad\square$

**Remark.** In contrast, residual-based methods (e.g., PINNs) may enter regions of parameter space where $\mathcal{L}u_\theta \not\approx 0$, resulting in sharp gradients, oscillatory behavior, or physically invalid outputs. Such instability is particularly problematic in stiff systems or ill-conditioned geometries.

**Conclusion.** By construction, NSNs constrain training dynamics to a physically valid subspace. As a result, optimization operates within a well-behaved manifold and avoids PDE-infeasible directions—resulting in smoother loss landscapes, more reliable convergence, and interpretable intermediate solutions.

