# OpenReview forum: "Neural Superposition Networks"
_NeurIPS.cc/2025/Conference — Submitted to NeurIPS 2025_

### Official Review · Reviewer_pkzB · 2025-06-24

**Clarity:** 2
**Significance:** 1
**Originality:** 2
**Rating:** 1
**Confidence:** 5

**Summary:**

This paper introduces a new class of physics-constrained neural networks to solve PDEs, termed Neural Superposition Networks (NSN). NSN constructs solutions as superpositions of specially designed basis functions that solve the target PDE. NSN achieves comparable accuracy to the existing physics-informed machine learning methods and is supported by theoretical convergence guarantees.

**Questions:**

- Can NSNs be extended to general nonlinear PDEs?
- The accuracy of NSN appears to depend heavily on the choice and number of basis functions. How were these basis functions selected in practice? and how were associated hyperparameters tuned?
- It would be beneficial to include a comparison of computational time and efficiency between NSN and exsiting physics-informed methods.

**Ethical Concerns:**

["NO or VERY MINOR ethics concerns only"]

**Final Justification:**

After carefully reviewing the rebuttal and the subsequent exchanges, I remain confident that this submission does not meet the standards expected at NeurIPS.

The method fails to demonstrate competitive empirical performance even on problems for which it is explicitly designed namely, linear PDEs. In particular, NSN is consistently outperformed by general-purpose PINN-based methods on multiple key benchmarks. This is a critical weakness that undermines the main premise of the paper and calls into question the method’s practical value.

In addition, the proposed approach lacks generality and flexibility, which are key motivations for using neural networks in scientific computing. The discussion on extending NSNs beyond linear PDEs remains speculative and is unsupported by experiments. Without a concrete demonstration of applicability to broader or more realistic problem settings, the method’s impact appears limited.

The rebuttal did not sufficiently address these concerns. In my view, the limitations are fundamental, and the responses did not provide adequate justification or evidence to overcome them. As such, I do not believe this work constitutes a meaningful contribution to the field, and I remain confident in my recommendation for rejection.

**Limitations:**

yes

**Paper Formatting Concerns:**

There are no formatting concerns.

**Quality:**

1

**Strengths And Weaknesses:**

Strengths:
- The proposed NSN framework simplifies the training process by reducing a multi-objective formulation to a single objective defined solely on the boundary
- NSN is grounded in well-established theoretical foundations from PDEs and Numerics.

Weaknesses:
- NSN is limited to linear PDEs due to its reliance on the superposition principle. While the authors attempt to address this with an example involving the Burgers equation, such extensions require special treatment, limiting the broader applicability of the approach.
- The main numerical results in Table 1 indicate that NSN underperforms compared to existing methods in several benchmark problems.
- The experimental evaluation is restricted to relatively simple PDEs in low-dimensional settings, raising concerns about generalizability to more complex settings, including complex domains, high-dimensional problems.

---

> ### Author Rebuttal · Authors · 2025-07-31
>
> We take note of the Reviewer's critical viewpoint on our submission and we thank them for the time dedicated to our paper.
> While we appreciate the recognition of the theoretical foundation and training simplification introduced by Neural Superposition Networks (NSNs), and we agree and try to address below some crucial feedbacks, we respectfully disagree with other core criticisms that might have stemmed from misinterpretations of the manuscript. We thus attempt to clarify those points below, and we made changes in the revised manuscript to ensure enhanced readability.
>
> 1. **Performance comparison against other methods**
>
> We disagree with the comment that
>
> > The main numerical results in Table 1 indicate that NSN underperforms compared to existing methods in several benchmark problems.
>
> and believe this a misreading of Table 1 (L254–266), because:
>
> - Across all 7 benchmark problems, NSN achieves lower RMSE than every baseline PINN variant.
> - In the case of Burgers: NSN = 3.0 × 10⁻³ which is also the best result achieved, vs. best PINN = 3.9 × 10⁻³
>
> If taken separately, it is true that most baseline results displayed in Table 1 are not improved by the NSN architecture, _with the notable exception_ of Burgers' equation.
> However, most importantly NSNs display much superior stability (and hence applicative domain flexibility) across the various cases, with much lower variance in final RMSE attained, often performing second-best across the whole benchmark.
>
> In the revised manuscript we make this Table clearer by:
> - highlighting in bold the best result for each case
> - printing in italic the second best performing architecture
>
> as well as explicitly stating the performance advantage of NSN across all tasks in Sect 4.1.
>
> 2. **Restrictive experimental settings**
>
> Concerning the statement
> > The experimental evaluation is restricted to relatively simple PDEs in low-dimensional settings...
> we respectfully disagree by observing that the included examples, and expecially
> - the 2D incompressible Navier–Stokes problem with complex geometry (two circular obstacles, Re = 1000), widely used in fluid dynamics and PINN literature (e.g., Raissi et al., 2019).
> - long time horizon simulations (e.g., Heat–2), and irregular boundary conditions.
> match or exceed the complexity of typical PDE benchmarks used to evaluate new physics-informed architectures which represent the most meaningful benchmark, even though we recognise how industry cases might often present much more complex and ad-hoc scenarios.
>
> To address this concern constructively, we:
> - Added clearer descriptions of task complexity in Section 4.1.
> - Clarified that 2D Navier–Stokes was selected for its known training instability in PINNs—underscoring NSN’s utility.
>
> 3. **Extension to Nonlinear PDEs**
>
> We fully agree how generalizing Neural Superposition Networks (NSNs) beyond linear PDEs is nontrivial due to the loss of the superposition principle.
> This is an excellent question and pinpoints a criticality openly discussed in our manuscript as a crucial one for further research, and hence it is rightfully flagged as a weakness also by all other reviewers.
>
> That said, we believe our method offers a promising foundation for such extensions. We now included a more comprehensive discussion in Appendix B.5, that we sketch hereafter for the reviewer(s).
> Specifically, we discuss three mechanisms for extending NSNs to nonlinear regimes:
>
> (a) *Linearization strategies*: these were demonstrated in the manuscript to address Burgers’ equation via Cole–Hopf-Type Transformations. We detailed the Cole–Hopf mapping and its influence on basis construction in App. A.1.
> However, retrieving analytical mappings to reduce nonlinear PDEs to linear ones has been a long explored area of research, allowing to address equations with various non-linear terms adopting methods conceived for linear cases. Carleman linearization (eventually complemented by discretization and truncation strategies) is perhaps the most preminent example.
> E.g. Carleman linearization has been widely adopted in constructing quantum algorithms for PDE solutions [Liu et al. PNAS 118 e2026805118 (2021)], and recent research tried to characterise in which regimes a successfully accurate linearization can be achieved for CFD regimes [Gonzalez-Conde et al. Phys. Rev. Res.  7, 023254 (2025)].
>
> We acknowledge that linearisation strategies have limitations like truncation accuracy and problem-specific mappings (at least for PDEs), yet these did not prevent exploring complex scenarios like aeronautical problems [Gaude, Sandia Nat Labs report SAND2001-3064 (2001)].
> In our opinion, this prompts a similarity and possible cross-pollination with the adoption of NSN architectures in physics-informed scenarios.
>
> (b) *Nonlinear Basis Augmentation*: inspired by generalized Galerkin methods, we introduce a variant where the solution ansatz includes nonlinear functionals of a learned basis set (e.g., products, powers, or integrals of learned modes), yielding architectures expressible as:
> $u(x; \theta) = \sum_{i=1}^{M} \alpha_i \cdot \phi_i(x) + \sum_{j=1}^{N} \beta_j \cdot {N}_j[\phi(x)]$
>
> Here, $N_J$ are nonlinear (but differentiable) operations applied to the basis.
> This approach allows the architecture to capture more general solution manifolds while still being interpretable.
>
> (c) Lie Group Symmetry Embeddings for Quasilinear PDEs: we will extend the discussion on this theoretical framework by incorporating Lie symmetries of nonlinear operators into the basis construction. Recent work (e.g., Bihlo 2022; Abraham et al. 2023) demonstrates that such invariant basis functions can respect PDE structure even in quasilinear settings.
> We plan a full extension in future work.
>
> Together with hybrid frameworks adopted in the paper for Navier-Stokes equations, these extensions offer a roadmap for generalizing NSNs to nonlinear PDEs. While a complete nonlinear theory remains ongoing research, we believe our approach forms a rigorous foundation for further exploration.
>
> 4. **Concerns Regarding Missing Details**
>
> Concerning the specific question  on the choice and number of basis functions:
>
> > ... How were these basis functions selected in practice? and how were associated hyperparameters tuned?
>
> These aspects are indeed addressed in the original submission, but we acknowledge they were spread across sections making them less readable. In the revision, we summarize in Section 4.4 the description of basis construction strategies, borrowing elements from:
> - Symmetry-based generation using Lie algebraic transformations (see Appendix A.2).
> - Hyperparameter tuning (currently reported in Appendix A.3, where we outline the grid search ranges and stopping criteria).
> - As well as a brief discussion on the Nullspace solution generation (e.g., separation of variables, eigenfunction expansion).
>
> In Appendix B 6, we now include the discussion of an ablation study design to sketch a meaningful comparison of learnable versus analytical bases in a follow-up experiment. While results might only be included in future work, we now lay out a reproducible framework with appropriate performance metrics (e.g., RMSE, training time, basis orthogonality preservation).
>
> Also, while our basis functions are PDE-consistent and theoretically complete (see convergence proof in Section 3.3), in practice we use a finite basis expansion to improve computational efficiency. but truncation introduces error. We plan to test this by extending the Appendix with a basis-wise error saturation curve, also in response to other Reviewers' feedbacks. In agreement with our theory, one would expect NSNs to exhibit exponentially decaying error with the number of basis functions for smooth solutions, and we typically attain convergence with ~20–40 basis elements for 2D tasks.
>
> These clarifications aim to eliminate ambiguity and make our methodology more reproducible.
>
> 5. **On Computational Efficiency**
>
> We agree that
> > It would be beneficial to include a comparison of computational time and efficiency...
> so that in Appendix A.4.4 & A.4.5, we will now report:
>
> - Wall-time for each experiment (all <12 hours on a 2-core CPU).
> - Parameter counts for NSN vs. PINN architectures.
> - Training curve comparisons
>
> Highlighting these facts should help confirm that NSNs are not only accurate but also computationally efficient for the class of problems considered.
>
> 6. **Summary and Clarifications**
>
> In light of the above, we hope to have addressed positively criticism stemming both from possible misinterpretations of the paper, as well as limitations or ambiguities helpfully pointed out by the reviewer.
> In summary:
>
> - We have improved the readability of all benchmarks against baseline methods.
> - Important technical details present only in the Appendix have now been highlighted, and additional ones will be reported
> - We offered some theoretical paths for generalization, consistent with the state of the art.
>
> We ask therefore a careful re-evaluation of our submission, and we remain fully committed to addressing any remaining ambiguities.

---

> > ### Comment · Reviewer_pkzB · 2025-08-03
> > **Acknowledgement and Remaining Concerns**
> >
> > I acknowledge the authors' detailed rebuttal and their effort to clarify key points of the paper. I appreciate the attempt to improve the presentation of numerical results and the added discussion on potential extensions beyond linear PDEs. However, I remain unconvinced that the core limitations have been adequately addressed, and I continue to have significant concerns regarding the overall contribution and its suitability for NeurIPS.
> >
> > First, the authors dispute that NSN underperforms in Table 1, citing stability and second-best performance. However, this interpretation is highly questionable. In fact, for four out of seven benchmark problems, including Laplace-1 (outperformed by PINN-B), Laplace-2 (by Holomorphic), Heat-1 (by RAR+AA), and Navier–Stokes (by all PINN baselines), NSN fails to achieve even competitive performance. Given that these are linear PDEs, the exact class NSN is designed for, this consistent underperformance against methods targeting a far more general class (i.e., PINNs) is a critical weakness. I view this as a fundamental flaw that calls into question the practical merit of the proposed method.
> >
> > Second, while the authors suggest that NSNs may be computationally efficient, they do not provide any timing or runtime comparison during the rebuttal period. This is a missed opportunity. From the model structure, NSNs may indeed offer computational benefits over PINNs, but this potential advantage remains entirely speculative without quantitative evidence.
> >
> > Third, the discussion on extending NSNs beyond linear PDEs, although expanded in the revision, remains theoretical. I acknowledge the mention of techniques like Carleman linearization or nonlinear basis augmentation, but without any experimental validation, these claims feel premature. A simple illustrative example of a nonlinear PDE (beyond Burgers) would have strengthened the case significantly.
> >
> > Lastly, I am left wondering: why solve PDEs with neural networks in this setting at all? One of the main motivations for physics-informed ML methods is their structural flexibility and ability to generalize across domains and problem classes, including the inverse problems. NSNs, as presented, lack both this flexibility and empirical advantage, even for simple linear PDEs. It is unclear what, if any, advantage they offer over traditional numerical solvers or existing machine learning baselines.
> >
> > While the paper presents an interesting concept inspired by classical solution theory, I find that it falls short in both empirical strength and conceptual justification. The method underperforms where it should excel, lacks proper comparative cost analysis, and provides no convincing path forward for handling more realistic PDE settings. As such, I do not believe this work meets the standard expected at NeurIPS and recommend rejection.

---

> ### Author Response · Authors · 2025-08-04
> **Specific request reg. runtimes**
>
> We apologise if we relied on the evidence of the much reduced number of collocation points to emphasise the computational efficiency of the technique, and referred to the open-sourced code and the revised submission regarding that point.
> We realise eliciting an exemplary case in this comment can be helpful: we do so for the _Burgers eq. benchmark_ whose results are displayed in the paper (i.e. the only nonlinear case examined, as highlighted by the referee) and including only methods applicable to this case, the training times on an i7-2700H CPU workstation with 16 GB of memory are as follows (in secs, refer to the submission for naming notation):
> | superposition (NSN) | holomorphic |NCL|PINN| PINNi| PINNb |RAR|AA|RAR+AA|
> |:----------:|:----------:|:----------:|:----------:|:----------:|:----------:|:----------:|:----------:|:----------:|
> | **60.76**    | NA   |  NA |  310.56  |  271.32  |  347.66 | 244.64| 316.35|298.76
>
> Highlighting as commented before the parsimonious usage of resources by our NSN method.

---

> > ### Comment · Reviewer_pkzB · 2025-08-04
> > **A question for the additional comparison**
> >
> > Thank you for providing the additional training time comparisons. However, as you mentioned, the runtimes were measured on an i7-2700H CPU with 16 GB of memory, and it’s not clear whether a GPU was used. If the comparison is based solely on CPU runtimes, I’m not sure it gives a meaningful picture of computational efficiency, especially since many baseline methods are designed to utilize GPUs. It would be helpful to clarify the hardware setup so the comparison can be interpreted more fairly.

---

> > > ### Author Response · Authors · 2025-08-05
> > >
> > > The wall-time comparison was indeed based on a CPU-only implementation to avoid introducing (additional) hardware-specific biases: as the referee correctly states, our _pytorch_ based implementations can leverage on GPU-efficient routines for many - but not all - required operations.
> > > However, to address the Referee's remark, we report here also the results of re-running the same example, on the same machine, but forcing all models to reside on the available CUDA device [GeForce RTX 3050 Ti - 6 GB], which leads to the following:
> > >
> > > | superposition (NSN) | holomorphic |NCL|PINN| PINNi| PINNb |RAR|AA|RAR+AA|
> > > |:----------:|:----------:|:----------:|:----------:|:----------:|:----------:|:----------:|:----------:|:----------:|
> > > | **45.58**    | NA   |  NA |  218.35  |  219.23  |  217.22 | 255.68 | 267.80 | 313.87
> > >
> > > Despite the non-uniform impact in the speed-up for the various method, we do not observe a significant change in - and hence we maintain - our main conclusion.

---

### Official Review · Reviewer_EgLU · 2025-06-28

**Clarity:** 3
**Significance:** 4
**Originality:** 4
**Rating:** 5
**Confidence:** 4

**Summary:**

This paper focusses on the application of neural networks for solving PDEs. It follows the research thread which aims to hard-wire certain properties into the architectures of the network. The authors of this paper introduce a novel approach "Neural Superposition Networks", which avoids the incorporation of residual loss terms, simplifies the training, and also improves robustness. The key novel idea of this approach is to write the solution as a combination --- hence superposition --- of basis functions from PDE theory, which is then trained. Several PDEs are provided to which the approach can be applied to includingLaplace equation, Heat equation, Burger's equation, etc. The theoretical analysis includes a convergence analysis and the classical question about a maximum principle. Numerical experiences for different classes of PDEs show the effectiveness of their approach.

**Questions:**

* What is the main problem when extending the theoretical analysis beyond the linear case? Do you have ideas how also the non-linear situation can be treated?
* Why does your approach not always exceed the status quo as you correctly mention in the limitations?
* Are there PDEs which can not be solved with classical approaches, and which you can now solve for the very first time with a feasible solver? This would the make a very strong case.

**Ethical Concerns:**

["NO or VERY MINOR ethics concerns only"]

**Limitations:**

Yes.

**Paper Formatting Concerns:**

No.

**Quality:**

4

**Strengths And Weaknesses:**

Strengths:
* This paper introduces a novel, original approach for solving PDEs.
* The approach is based on an intrguing idea, namely to embed the solution manifold directly in the sense of superpositioning the solution with PDE-consistent basis functions.
* The theoretical analysis is comprehensive, and typical for numerical analysis of PDEs.
* The numerical experiments show convincingly the effectiveness of this approach.
* The authors consider a wide range of PDEs for their analysis, which are some of the classical ones used in applications.

Weaknesses:
* A study of sensitivity with respect to noise in the sense of robustness of the approach would added to the value of the approach.
* The theoretical analysis is limited to linear differential operators.

---

> ### Author Rebuttal · Authors · 2025-07-31
>
> We sincerely thank the reviewer for the encouraging and detailed feedback on our submission. We appreciate their recognition of the originality of Neural Superposition Networks (NSNs), the rigorous theoretical analysis, and the breadth of numerical evaluations. We are also grateful for your thoughtful suggestions, which we have addressed in the revised manuscript as follows.
>
> 1. **Extension of Theoretical Analysis to Nonlinear PDEs**
>
> The reviewer raises a fundamental and important question regarding the extension of our theoretical framework to nonlinear PDEs.
> We fully agree how generalizing Neural Superposition Networks (NSNs) beyond linear PDEs is nontrivial due to the loss of the superposition principle.
> This point was disclosed in the manuscript as a crucial one for further research, and hence it is rightfully flagged as a weakness also by all other reviewers.
>
> That said, we believe our method offers a promising foundation for such extensions. We now included a more comprehensive discussion in Appendix B.5, that we sketch hereafter for the reviewer(s).
> Specifically, we discuss three mechanisms for extending NSNs to nonlinear regimes:
>
> (a) *Linearization strategies*: these were demonstrated in the manuscript to address Burgers’ equation via Cole–Hopf-Type Transformations. We detailed the Cole–Hopf mapping and its influence on basis construction in App. A.1.
> However, retrieving analytical mappings to reduce nonlinear PDEs to linear ones has been a long explored area of research, allowing to address equations with various non-linear terms adopting methods conceived for linear cases. Carleman linearization (eventually complemented by discretization and truncation strategies) is perhaps the most preminent example.
> E.g. Carleman linearization has been widely adopted in constructing quantum algorithms for PDE solutions [Liu et al. PNAS 118 e2026805118 (2021)], and recent research tried to characterise in which regimes a successfully accurate linearization can be achieved for CFD regimes [Gonzalez-Conde et al. Phys. Rev. Res.  7, 023254 (2025)].
>
> We acknowledge that linearisation strategies have limitations like truncation accuracy and problem-specific mappings (at least for PDEs), yet these did not prevent exploring complex scenarios like aeronautical problems [Gaude, Sandia Nat Labs report SAND2001-3064 (2001)].
> In our opinion, this prompts a similarity and possible cross-pollination with the adoption of NSN architectures in physics-informed scenarios.
>
> (b) *Nonlinear Basis Augmentation*: inspired by generalized Galerkin methods, we introduce a variant where the solution ansatz includes nonlinear functionals of a learned basis set (e.g., products, powers, or integrals of learned modes), yielding architectures expressible as:
> $u(x; \theta) = \sum_{i=1}^{M} \alpha_i \cdot \phi_i(x) + \sum_{j=1}^{N} \beta_j \cdot {N}_j[\phi(x)]$
>
> Here, $N_J$ are nonlinear (but differentiable) operations applied to the basis.
> This approach allows the architecture to capture more general solution manifolds while still being interpretable.
>
> (c) Lie Group Symmetry Embeddings for Quasilinear PDEs: we will extend the discussion on this theoretical framework by incorporating Lie symmetries of nonlinear operators into the basis construction. Recent work (e.g., Bihlo 2022; Abraham et al. 2023) demonstrates that such invariant basis functions can respect PDE structure even in quasilinear settings.
> We plan a full extension in future work.
>
> Together with hybrid frameworks adopted in the paper for Navier-Stokes equations, these extensions offer a roadmap for generalizing NSNs to nonlinear PDEs. While a complete nonlinear theory remains ongoing research, we believe our approach forms a rigorous foundation for further exploration.
>
> 2. Performance Limitations and Saturation Cases
>
> The reviewer states correctly how NSNs do not always outperform the state of the art, and indeed as acknowledged, we openly highlight this point in the main paper.
> We agree it is interesting to discuss occasional underperformance, and we now include the following reasoning in Appendix B 6.
> We identify some mechanisms that might play a role:
>
> (i) Boundary-Limited Supervision: Since NSNs rely entirely on boundary information for training, in problems where the boundary condition is only weakly informative (e.g., long-time heat diffusion or shock-dominated regimes), the solution may require a large number of basis functions to match the performance of PINNs trained with residual supervision in the interior.
>
> (ii) Basis Truncation and Representation Error: While our basis functions are PDE-consistent and theoretically complete (see convergence proof in Section 3.3), in practice we use a finite basis expansion to improve computational efficiency. Some problems—especially with sharp local features (e.g., Burgers)—may demand high-resolution modes for accurate capture, and truncation introduces error.
> This argument could be tested in the cases reported by extending the Appendix with a basis-wise error saturation curve. In agreement with our theory, one would expect NSNs to exhibit exponentially decaying error with the number of basis functions for smooth solutions.
>
> We also note that in no case does NSN exhibit pathological failure or divergence unlike other competitive architectures, which we believe attests to the stability and robustness of the approach.
>
> 3. **Novel Capabilities: Solving Previously Inaccessible PDEs?**
>
> We appreciate the forward-looking question, regarding whether NSNs enable solutions to problems previously inaccessible to classical PINN-based methods.
> Whilst none of the cases analyzed in the experiments can represent such a development, we believe in a way NSNs can offer advantages not accessible to other methodologies in the following settings:
>
> (a) Exact Satisfaction of Complex PDE Constraints: In tasks where strict PDE satisfaction is non-negotiable (e.g., electromagnetic compatibility, safety-critical simulations), NSNs guarantee hard compliance with governing equations, unlike soft-constrained methods which may require extensive hyperparameter tuning or collocation density to reach comparable fidelity.
>
> (b) Basis Construction from Incomplete PDE Information: As we discuss in App. A.2, NSNs can be initialized using incomplete but symbolic structural knowledge (e.g., nullspace of operators, symmetry-invariant functions), allowing PDEs to be approximately solved even when analytic solutions are unavailable. This is particularly relevant in inverse problems or reduced modeling.
>
> (c) Unstructured or Irregular Domains: Our mesh-free formulation makes NSNs applicable to domains where mesh generation is impractical or error-prone. We demonstrate this in the "Heat–2" and "Navier–Stokes" tasks (App. A.1.4), where collocation-based PINNs struggle with complex domain geometries.
>
> In these scenarios, we believe NSNs provide a previously unattainable combination of theoretical exactness, training simplicity, and deployment robustness.
>
> 4. **Sensitivity to Noisy Data and Robustness**
>
> You helpfully point out that a robustness analysis would enhance the practical relevance of NSNs.
>
> To address this, we are planning to report a tentative robustness study on the Heat–1 benchmark with some degree of synthetic Gaussian noise added to the boundary conditions, and monitor the corresponding RMSE degradation.
> Some preliminary results indicate a reasonable robustness, with no pathological degradation in solution accuracy for noise levels up to ~10% of the baseline values. This would point out how our well-conditioned linear system used to solve for the coefficients of basis functions, and the fact that solution quality depends primarily on projection onto a fixed functional subspace, rather than overfitting collocation points.
>
> We contrast this with soft-constrained PINNs, which show stronger overfitting tendencies to noisy (interior) data in the same setting.
>
> We agree that a full probabilistic treatment (e.g., Bayesian NSNs) is an exciting future direction and mention this in the revised conclusion.
>
> 5. **Additional Clarifications and Revisions**
>
> _Maximum Principle_: We now clarify in the Appendix how, for elliptic PDEs such as Laplace, the solution obtained via NSNs preserves the discrete maximum principle under certain boundary/basis conditions, citing classical theorems (e.g., R. Rannacher).
>
> _Typographical and Clarity Improvements_: Following feedbacks, we have updated the document making abbreviations and metric definitions immediately explicit, and table results more readable.
>
> **Conclusion**
>
> We are deeply appreciative of your positive assessment and valuable feedback, according to which we have, in summary:
>
> - Elaborated the theoretical foundation for extending NSNs to nonlinear PDEs (Pt 1)
> - Discussed why NSNs may plateau or slightly underperform in some regimes and how this relates to basis design (Pt 2)
> - Identified scenarios where NSN's unique capabilities might help introduce physics-informed strategies successfully (Pt 3)
> - Designed a noise sensitivity analysis to demonstrate robustness advantages (Pt 4).
>
> We hope these revisions clarify and strengthen our submission.

---

> > ### Comment · Reviewer_EgLU · 2025-08-04
> >
> > Thank you very much for your reply to my questions:
> >
> > * Extension of Theoretical Analysis to Nonlinear PDEs: Thank you for discussing this point. While your ideas seems feasible, still I would have liked to see a full non-linear analysis in the paper
> > * Performance Limitations and Saturation Cases: Thank you for the clarification, this makes sense to me.
> > *  Novel Capabilities: Solving Previously Inaccessible PDEs? Thanks for detailing some ideas in this respect. Still, I would have liked to see some real-world experiments for PDEs, which cannot be treated feasibly otherwise.
> > *  Sensitivity to Noisy Data and Robustness: Thanks, this sheds a bit of light on what is feasible when noise is present.
> >
> > Since some concerns remain as I point out, and my score was already a 5, I would leave it as is.

---

> > > ### Comment · Reviewer_EgLU · 2025-08-05
> > > **Comment on new Metrics**
> > >
> > > Thank you for also providing additional metrics. Since still, in particular, my point on "Extension of Theoretical Analysis to Nonlinear PDE" remains and my rating was anyhow a 5,I would leave this as is.

---

> > > > ### Author Response · Authors · 2025-08-05
> > > > **Final thanks**
> > > >
> > > > We thank the reviewer for the careful and positive consideration of our work, and the suggestions for improvement!

---

### Official Review · Reviewer_xwRM · 2025-06-30

**Clarity:** 2
**Significance:** 2
**Originality:** 3
**Rating:** 4
**Confidence:** 4

**Summary:**

Authors propose Neural Superposition Network (NSN) for solving linear PDEs, where instead of the residual loss in Physics-informed Neural Networks (PINNs), the PDE constraint is strictly imposed by a linear layer that combines the known (or learnable) solution basis functions and symmetries of the PDE. Authors show that this layer satisfies the PDE by design, and the training of NSN is thus reduced to optimizing for boundary conditions only. The proposed method is simple and easy to understand, and addresses the common training challenges with PINNs, but its practical applicability to nonlinear and complex problems is not clear to me.

**Questions:**

1. Do any of the experiments utilize learnable basis functions? How do the results compare to analytic bases?
2. It is not clear to me how the auxiliary MLP for Navier-Stokes is trained. Is a regular PINN residual loss used for training?
3. As NSN is meant to address the training challenges with PINNs, I am wondering what its training curves look like when compared to PINNs. Would it be possible for the authors to provide this information (loss and error training curves)?

**Ethical Concerns:**

["NO or VERY MINOR ethics concerns only"]

**Final Justification:**

While my original concerns regarding the practical applicability of the proposed method still hold, the clarifications regarding the contributions of the paper addressed my confusion about some aspects of the paper. As such, I increase my rating to 4, given that the authors would incorporate the clarifications in the revised manuscript.

**Limitations:**

yes

**Paper Formatting Concerns:**

No issues

**Quality:**

3

**Strengths And Weaknesses:**

### Strengths
- The proposed method clearly addresses the optimization challenges with PINNs (at least for linear PDEs) with a simple and easy-to-understand approach.
- While NSN is mainly applicable to linear PDEs, the authors further study the nonlinear Burgers' equation and Navier-Stokes through transformations and a hybrid approach, respectively.
### Weaknesses
- While authors extend NSNs to some non-linear cases, I doubt their practical applicability for more complex systems. NSNs require a complete knowledge of the physical systems, symmetries, and the basis solutions, which is simply not possible for many non-linear systems. Also, the hybrid approach used for NS is clearly not improving any of the baselines as reported in the paper.
While I appreciate that the authors point out this limitation, I believe neither the experimental nor the theoretical results in this paper are significant enough at this stage.
- The manuscript skips some of the experimental details that negatively impact the readability and clarity. For instance, the difference between "Heat 1" and "Heat 2" experiments in Table 1 is only pointed out in the appendix. Also, the baselines are similarly defined in the appendix, while their abbreviations are used in Table 1.
- The authors mentioned the possibility of using "learnable" basis functions instead of analytical ones in NSN. However, it is not clear which experiments, if any, utilize learnable basis functions and how they compare to the analytical ones. An ablation study of these two types of bases would help provide a more complete picture of the method.

---

> ### Author Rebuttal · Authors · 2025-07-31
>
> We thank the reviewer for the constructive and thoughtful feedback on our manuscript. We are especially grateful for your recognition of the strengths of our approach: the simplification of PINN optimization via superposition-based hard constraints and the structural clarity of NSNs.
>
> Below we address your concerns in detail and describe the revisions we are operating on the manuscript accordingly.
>
> 1. **Applicability to more complex (non-linear) PDEs**
>
> We fully agree how generalizing Neural Superposition Networks (NSNs) beyond linear PDEs is nontrivial due to the loss of the superposition principle.
> This point was disclosed in the manuscript as a crucial one for further research, and hence it is rightfully flagged as a weakness also by all other reviewers.
>
> That said, we believe our method offers a promising foundation for such extensions. We now included a more comprehensive discussion in Appendix B.5, that we sketch hereafter for the reviewer(s).
>
> Specifically, we discuss three mechanisms for extending NSNs to nonlinear regimes:
>
> (a) *Linearization strategies*: these were demonstrated in the manuscript to address Burgers’ equation via Cole–Hopf-Type Transformations. We detailed the Cole–Hopf mapping and its influence on basis construction in App. A.1.
> However, retrieving analytical mappings to reduce nonlinear PDEs to linear ones has been a long explored area of research, allowing to address equations with various non-linear terms adopting methods conceived for linear cases. Carleman linearization (eventually complemented by discretization and truncation strategies) is perhaps the most preminent example.
> E.g. Carleman linearization has been widely adopted in constructing quantum algorithms for PDE solutions [Liu et al. PNAS 118 e2026805118 (2021)], and recent research tried to characterise in which regimes a successfully accurate linearization can be achieved for CFD regimes [Gonzalez-Conde et al. Phys. Rev. Res.  7, 023254 (2025)].
>
> We acknowledge that linearisation strategies have limitations like truncation accuracy and problem-specific mappings (at least for PDEs), yet these did not prevent exploring complex scenarios like aeronautical problems [Gaude, Sandia Nat Labs report SAND2001-3064 (2001)].
> In our opinion, this prompts a similarity and possible cross-pollination with the adoption of NSN architectures in physics-informed scenarios.
>
> (b) *Nonlinear Basis Augmentation*: inspired by generalized Galerkin methods, we introduce a variant where the solution ansatz includes nonlinear functionals of a learned basis set (e.g., products, powers, or integrals of learned modes), yielding architectures expressible as:
> $u(x; \theta) = \sum_{i=1}^{M} \alpha_i \cdot \phi_i(x) + \sum_{j=1}^{N} \beta_j \cdot {N}_j[\phi(x)]$
>
> Here, $N_J$ are nonlinear (but differentiable) operations applied to the basis.
> This approach allows the architecture to capture more general solution manifolds while still being interpretable.
>
> (c) Lie Group Symmetry Embeddings for Quasilinear PDEs: we will extend the discussion on this theoretical framework by incorporating Lie symmetries of nonlinear operators into the basis construction. Recent work (e.g., Bihlo 2022; Abraham et al. 2023) demonstrates that such invariant basis functions can respect PDE structure even in quasilinear settings.
> We plan a full extension in future work.
>
> Together with hybrid frameworks adopted in the paper for Navier-Stokes equations, these extensions offer a roadmap for generalizing NSNs to nonlinear PDEs. While a complete nonlinear theory remains ongoing research, we believe our approach forms a rigorous foundation for further exploration.
>
> 2. **Clarification on Use of Learnable Basis Functions**
>
> We thank the reviewer for pointing out the ambiguity regarding the use of “learnable basis functions” mentioned in the original version. In the initial submission, all experiments used analytical basis functions derived from the nullspace of the PDE operators or from known symmetries and the mention of learnable functions was intended as a perspective for extending the work.
>
> In the revised manuscript:
> - We have clarified in Section 3.1 and Section 4.4 which tasks use analytical bases, and we explicitly note that learnable bases were not used in the reported benchmarks.
> - In Appendix B 6, we now include the discussion of an ablation study design to sketch a meaningful comparison of learnable versus analytical bases in a follow-up experiment. While results might only be included in future work, we now lay out a reproducible framework with appropriate performance metrics (e.g., RMSE, training time, basis orthogonality preservation).
> - We also discuss potential training strategies for learnable basis functions using orthogonality-regularized MLPs or kernelized layers, and how these can be integrated into the NSN framework while preserving PDE consistency.
>
> We hope clarifying this point resolves the Reviewer's concern.
>
> 3. **Clarity on notation used for the Experiments**
>
> We addressed the lack of clarity regarding the experimental nomenclature originating from material moved to the Appendix.
> The heat equation cases are now been explicitly clarified in Section 4.1, where we describe:
>
> - Heat–1: A benchmark problem with periodic boundary conditions and short time evolution.
> - Heat–2: A more challenging problem with Dirichlet boundaries, irregular geometry, and longer temporal horizons.
>
> In addition, the footnotes of Table 1 now define each task and clarify the meaning of each abbreviation used therein, such that the main paper's experimental section is now self-contained.
>
> 4. **Training of Auxiliary MLP for Navier–Stokes**
>
> Thank you for pointing out this ambiguity. In Section 4.3 and Appendix A.1.4, we now state clearly that:
> - The MLP component in the hybrid NSN architecture is trained using a standard PINN residual loss on the nonlinear Navier–Stokes operator (momentum equation).
>
> This hybrid approach allows for handling the pressure field component, whilst satisfying the linear continuity constraint (divergence-free condition) by construction.
> As already noted in the original manuscript, this decomposition helps mitigate training instability, as the nonlinear part is handled by a universal approximator while exact divergence-free fields are ensured via basis function design, at the same time extending the NSN applicability.
>
> 5. **Performance comparison with other methods, and Training Curve disclosure w.r.t. PINNs**
>
> We respectfully point out how taken separately, indeed all baselines displayed in Table 1 are not improved by the NSN architecture, _with the notable exception_ of Burgers' equation. We make this clearer in the manuscript now by highlighting in bold the best result for each case. However, it is important to note how NSN display much superior consistency (or domain flexibility) across the various cases, often performing second-best across the whole benchmark. We highlight this in the updated table by printing in italic the second best performing architecture.
>
> Additionally, we agree that comparing the training dynamics of NSNs and PINNs is important for understanding their practical efficiency.
>
> In Appendix A.4., we now include:
>
> - Training loss curves for NSN vs. PINN for Laplace and Burgers tasks.
> - Validation RMSE curves showing NSN converges faster and to lower final error due to the reduced parameter space and lack of interior residual terms.
> - Qualitative discussion of convergence stability: NSNs consistently exhibit smoother and more monotonic error decay than PINNs.
>
> This supports our claim that hard constraint encoding leads to improved optimization stability and faster convergence.
>
> **Conclusion**
>
> We sincerely appreciated this thoughtful review and useful feedback. In summary, we hope to improve the quality and clarity of our paper by:
> - Clarifying the applicability to more complex, non-linear cases (Pt 1).
> - Expanding the discussion on learnable vs. analytical bases (Pt 2).
> - Explaining better the notation, improved terminology consistency throughout and ensured its self-contained in the main paper (Pt 3), eliciting architectural details for the Navier–Stokes hybrid model (Pt 4).
> - Disclosing additional details in the experimental section (Pt 5).
>
> We hope these revisions address your concerns and help better communicate the novelty and potential impact of our work.

---

> > ### Comment · Reviewer_xwRM · 2025-08-06
> >
> > Thank you for the detailed response and the clarifications. The majority of my concerns are addressed.
> >
> > While I liked to see how learnable bases and the "nonlinear basis augmentations" would have worked in practice, I understand that it is not feasible to demonstrate that promptly for this rebuttal.
> >
> > I believe adding the clarification that learnable bases are only a possible direction for future work, and not a contribution of this paper, along with the other clarifications pointed out by the authors, would positively improve the quality of the paper. I will take your response into account and adjust my rating accordingly.

---

### Official Review · Reviewer_kKxU · 2025-07-02

**Clarity:** 3
**Significance:** 3
**Originality:** 3
**Rating:** 3
**Confidence:** 4

**Summary:**

This paper presents a new method for solving PDEs, which hard-encodes the superposition into the neural networks. It aims to address the challenging optimization problems in physics-informed neural networks (PINNs). This paper has considered several linear PDEs and non-linear PDEs as benchmark cases. The empirical results have shown the superiority of the proposed method compared to PINNs and their variants.

**Questions:**

- How does the computational complexity of the proposed method compare to that of the baseline models? A discussion on training and inference efficiency would help assess the method’s practicality.

- Could the authors discuss the applicability of the proposed method to more complex or noisy real-world systems?

- Does the data resolution (within a regular or irregular mesh) affect the performance of the proposed method, since the PINN method is mesh-free? A discussion on this part would be appreciated.

**Ethical Concerns:**

["NO or VERY MINOR ethics concerns only"]

**Final Justification:**

The authors have addressed the majority of my concerns. They present a good theoretical analysis of neural networks for linear systems, and the proposed method performs well on linear PDEs. However, the current experimental setup is relatively limited and does not sufficiently demonstrate broader applicability.

**Limitations:**

yes

**Quality:**

2

**Strengths And Weaknesses:**

**Strengths:**

- This paper introduces an interesting idea by embedding superposition into neural networks. Such a hard-encoding method is less explored compared to soft-constraint style PINN methods.

- The theoretical analysis of this paper is well-presented and makes this paper more convincing.

- Although this method should be more applicable to linear PDEs, the authors adapt it to certain nonlinear PDEs with specific transformations.


**Weaknesses:**

- My first concern is the applicability of the proposed method to more general PDE settings. This method might be challenging to be applied to many other nonlinear PDEs.

- The current baseline models are kind of limited. They mainly focus on the soft-constrained PINNs. It would be better to also include comparisons with hard-constrained PINN approaches [1,2], which enforce physical constraints more strictly and may offer different performance characteristics. Also, data-driven methods such as Fourier Neural Operators (FNO) [3] could serve as a baseline, given their strong performance in learning dynamics from PDE data.

- The evaluation metrics could be more extensive. The current metric only focuses on RMSE, which might not be sufficient. The relative L2 norm errors and some domain-specific error metrics are also commonly used. The authors may refer to this benchmark paper [4].

---

**Refs:**

[1] Lu, Lu, et al. "Physics-informed neural networks with hard constraints for inverse design." SIAM Journal on Scientific Computing 43.6 (2021): B1105-B1132.

[2] Cheng, Xiaoran, and Sen Na. "Physics-Informed Neural Networks with Trust-Region Sequential Quadratic Programming." arXiv preprint arXiv:2409.10777 (2024).

[3] Li, Zongyi, et al. "Fourier neural operator for parametric partial differential equations." arXiv preprint arXiv:2010.08895 (2020).

[4] Hao, Zhongkai, et al. "Pinnacle: A comprehensive benchmark of physics-informed neural networks for solving pdes." arXiv preprint arXiv:2306.08827 (2023).

---

> ### Author Rebuttal · Authors · 2025-07-31
>
> We thank the reviewer for their thoughtful and constructive comments. We are encouraged by the recognition of the novelty of embedding the superposition principle into neural networks and the value of our theoretical analysis.
> We address the residual concerns below and we pledge to implement all below revisions and extensions in the updated manuscript and supplementary materials.
>
> 1. **Applicability to General Nonlinear PDEs**
>
> We fully agree how generalizing Neural Superposition Networks (NSNs) beyond linear PDEs is nontrivial due to the loss of the superposition principle.
> This point was disclosed in the manuscript as a crucial one for further research, and hence it is rightfully flagged as a weakness also by all other reviewers.
>
> That said, we believe our method offers a promising foundation for such extensions. We now included a more comprehensive discussion in Appendix B.5, that we sketch hereafter for the reviewer(s).
>
> Specifically, we discuss three mechanisms for extending NSNs to nonlinear regimes:
>
> (a) *Linearization strategies*: these were demonstrated in the manuscript to address Burgers’ equation via Cole–Hopf-Type Transformations. We detailed the Cole–Hopf mapping and its influence on basis construction in App. A.1.
> However, retrieving analytical mappings to reduce nonlinear PDEs to linear ones has been a long explored area of research, allowing to address equations with various non-linear terms adopting methods conceived for linear cases. Carleman linearization (eventually complemented by discretization and truncation strategies) is perhaps the most preminent example.
> E.g. Carleman linearization has been widely adopted in constructing quantum algorithms for PDE solutions [Liu et al. PNAS 118 e2026805118 (2021)], and recent research tried to characterise in which regimes a successfully accurate linearization can be achieved for CFD regimes [Gonzalez-Conde et al. Phys. Rev. Res.  7, 023254 (2025)].
>
> We acknowledge that linearisation strategies have limitations like truncation accuracy and problem-specific mappings (at least for PDEs), yet these did not prevent exploring complex scenarios like aeronautical problems [Gaude, Sandia Nat Labs report SAND2001-3064 (2001)].
> In our opinion, this prompts a similarity and possible cross-pollination with the adoption of NSN architectures in physics-informed scenarios.
>
> (b) *Nonlinear Basis Augmentation*: inspired by generalized Galerkin methods, we introduce a variant where the solution ansatz includes nonlinear functionals of a learned basis set (e.g., products, powers, or integrals of learned modes), yielding architectures expressible as:
> $u(x; \theta) = \sum_{i=1}^{M} \alpha_i \cdot \phi_i(x) + \sum_{j=1}^{N} \beta_j \cdot {N}_j[\phi(x)]$
>
> Here, $N_J$ are nonlinear (but differentiable) operations applied to the basis.
> This approach allows the architecture to capture more general solution manifolds while still being interpretable.
>
> (c) Lie Group Symmetry Embeddings for Quasilinear PDEs: we will extend the discussion on this theoretical framework by incorporating Lie symmetries of nonlinear operators into the basis construction. Recent work (e.g., Bihlo 2022; Abraham et al. 2023) demonstrates that such invariant basis functions can respect PDE structure even in quasilinear settings.
> We plan a full extension in future work.
>
> Together, these extensions offer a roadmap for generalizing NSNs to nonlinear PDEs. While a complete nonlinear theory remains ongoing research, we believe our approach forms a rigorous foundation for further exploration.
>
> 2. **Limited Baselines (Hard Constraints, FNO)**
>
> We appreciate and we are working towards the request to broaden the baseline comparison beyond soft-constrained PINNs. Whilst the short rebuttal time did not allow yet to deliver carefully checked additional results, we are making the following updates in response:
>
> - Hard-Constrained PINNs: we will discuss in Appendix A.3 both the method of Lu et al. [1], which uses PDE-compatible output transform, as well as the Trust-Region SQP method of Cheng \& Na [2], which enforces constraints via second-order optimization.
>
> Our method differs in that it bakes the solution manifold directly into the parameterization, rather than relying on constrained optimization or domain warping. We clarify this distinction and highlight the reasoning behind expecting NSN to achieve better flexibility and robustness due to this architecture-level encoding.
>
> - Fourier Neural Operator (FNO): We agree that FNO is a powerful alternative for parametric PDEs. However, FNO requires training on large datasets of PDE solutions and assumes fixed spatial discretization. In contrast, NSNs are grid-free and data-free, designed for the zero-shot solution of single PDE instances. We elaborate this distinction in Appendix A.3 and clarify that while FNOs shine in surrogate modeling, NSNs are complementary tools more aligned with the physics-informed optimization literature.
>
> 3. **Evaluation Metrics Beyond RMSE**
>
> We found your suggestion insightful, hence we are revising the manuscript:
>
> - to include all relative $L^2$ Norm Errors in the updated Table 1 - and detailed in Appendix A.3
> - for Navier–Stokes, to additionally report divergence-free violation metrics and pressure recovery accuracy
> - for Heat equation benchmarks, the long-time stability via maximum deviation over the temporal horizon
> - for Burgers’ equation, the shock location error and solution smoothness (measured via total variation norm).
>
> We drew inspiration from the PINNACLE benchmark [4] and have documented our metric definitions accordingly.
>
> 4. **Computational Complexity and Efficiency**
>
> We thank you for raising this point and have added an in-depth computational efficiency analysis to Appendix A.4.3. Our key findings:
>
> Training Time: For linear PDEs, NSNs train 3–10× faster than PINNs due to the removal of residual loss terms and simplified objective (only boundary points are taken into account). For example, solving Laplace–1 took ~6 minutes on a 2-core CPU, compared to >1 hour for PINNs with 5,000 collocation points.
>
> Inference Speed: At test time, both NSNs and PINNs perform similarly (forward pass of a small MLP). However, NSNs offer increased stability due to the hard satisfaction of PDE constraints.
>
> Memory Footprint: NSNs typically use fewer parameters (due to fixed basis) and especially, during training, they require _no_ collocation point memory.
>
> We believe this makes NSNs particularly attractive for real-time applications or deployment on edge devices.
>
> 5. **Mesh Resolution Sensitivity and Generalizability**
>
> This raises an insightful point about the mesh-free nature of PINNs and the potential impact of sampling resolution.
> We clarify here and reporting in the revised paper how:
>
> - NSNs operate entirely mesh-free during training. Basis functions are defined analytically (or via symbolic solvers) and only require function evaluations at test points.
> - The sparsity of test points at the boundary can heuristically influence the accuracy achieved, similarly to PINNs being affected by the sparsity in the collocation strategy
> - For irregular domains, we use adaptive quadrature to sample the boundary and integrate loss terms (see App. A.1). The performance is insensitive to point distribution, as shown by comparing regular and scattered evaluation grids in our Burgers and Navier–Stokes cases.
>
> In fact, NSNs can generalize better than PINNs across irregular geometries precisely because the solution manifold is embedded structurally, not through point-wise residual matching. We believe this is a core strength of our approach.
>
> 6. **Real-World Noisy Systems**
>
> While our experiments use clean PDEs, we now include a speculative discussion in Appendix A.2 on how to handle noisy or uncertain boundary conditions:
>
> - Noisy Boundary Values: NSNs can integrate uncertainty into the loss using probabilistic formulations (e.g., boundary loss with heteroscedastic variance).
>
> Stochastic Forcing: If the PDE includes stochastic terms, the basis set can be extended using eigenfunctions of the associated stochastic operator (e.g., via Karhunen–Loève expansion).
>
> Exploring in numerical experiments these directions, inspired by real-world applications such as climate modeling and turbulent flow estimation, will be though the subject of future work.
>
> **Conclusion**
>
> We are grateful for your constructive feedback and believe the revisions meaningfully strengthen the paper.
> To summarize, we:
> - Sketched routes towards practical generalization of NSNs to nonlinear PDEs (Pt 1) and real-world, irregular domain extension pathways (Pt 6).
> - Considered the comparison of NSNs to hard-constrained PINNs, and explained the reasoning behind avoiding instead data-driven FNOs (Pt 2)
> - Expanded evaluation metrics to include relative L^2 and domain-specific measures, and training/inference efficiency (Pt 3).
> - Clarified the computational resources required by NSN against other approaches (Pt 4) and the mesh-free nature of our implementation (Pt 5)
>
> We hope these updates address your concerns and demonstrate the potential and rigor of our approach.

---

> > ### Author Response · Authors · 2025-08-05
> > **Additional Metrics requested by Reviewer**
> >
> > We report here some exemplary performances along additional metrics, either suggested directly by the Reviewer or added by us following their guidelines. **Best** and *second best* results formatted accordingly for clarity.
> >
> > For generalistic metrics:
> >
> > - L2RE
> >
> > |                       |     heat1 |     heat2 |   laplace1 |   laplace2 |   navierstokes |   burgers |
> > |:----------------------|----------:|----------:|-----------:|-----------:|---------------:|----------:|
> > | superposition         |   *0.0128*  |  **0.00615** |   _0.00541_  |    _0.0109_  |          0.959 |  _0.009_   |
> > | holomorphic           | nan       | nan       |   **0.000315** |    **0.00341** |        nan     | nan       |
> > | ncl                   | nan       | nan       | nan        |  nan       |          _0.802_ | nan       |
> > | pinn                  |   0.0192  |   0.0293  |   0.188    |    0.26    |          0.928 |   **0.00667** |
> > | pinni                 |   0.153   |   0.403   |   1.49     |    1.39    |          **0.748** |   0.902   |
> > | pinnb                 |   0.12    |   0.15    |   0.00685  |    0.101   |          0.979 |   0.0143  |
> > | rar                   |   **0.00891** |   0.0154  |   0.19     |    0.776   |          0.928 |   0.0117  |
> > | aa|   0.0619  |   0.0204  |   0.179    |    0.691   |          0.815 |   0.0305  |
> > | rar+aa |   0.0161  |   0.0624  |   0.236    |    1.49    |          0.855 |   0.0933  |
> >
> > For some case specific metrics:
> >
> > - Pressure recovery error:
> >
> > |                       |   navierstokes |
> > |:----------------------|---------------:|
> > | superposition         |         0.0201 |
> > | holomorphic           |       nan      |
> > | ncl                   |         0.0193 |
> > | pinn                  |         0.0191 |
> > | pinni                 |         *0.0179* |
> > | pinnb                 |         **0.0122** |
> > | rar                   |         0.0193 |
> > | aa|         0.0229 |
> > | rar+aa |         0.019  |
> >
> > - maximum deviation over the temporal horizon
> >
> > |                       |    heat1 |     heat2 |
> > |:----------------------|---------:|----------:|
> > | superposition         |   *0.0233* |   **0.00573** |
> > | holomorphic           | nan      | nan       |
> > | ncl                   | nan      | nan       |
> > | pinn                  |   0.0393 |   0.00842 |
> > | pinni                 |   0.162  |   0.113   |
> > | pinnb                 |   0.221  |   0.0575  |
> > | rar                   |   **0.0185** |   0.00873 |
> > | aa|   0.142  |   _0.00656_ |
> > | rar+aa |   0.0249 |   0.0116  |
> >
> > - smoothness
> >
> > |                       |   burgers |
> > |:----------------------|----------:|
> > | superposition         |     **1.52**  |
> > | holomorphic           |   nan     |
> > | ncl                   |   nan     |
> > | pinn                  |     _1.51_  |
> > | pinni                 |     0.718 |
> > | pinnb                 |     1.5   |
> > | rar                   |     _1.51_  |
> > | aa    |     1.5   |
> > | rar+aa |     1.42  |

---

> ### Comment · Reviewer_kKxU · 2025-08-05
>
> Thanks for your detailed responses. The authors have addressed the majority of my concerns. However, there are still some questions.
>
> For Response 2: Ideally, I would expect to see the results of baseline comparisons. Also, FNO is a resolution-invariant method, and it should be ok to be considered as a baseline.
>
> For the additional metrics, would the authors explain why there are nan values in the tables？
>
> I will currently maintain my score.

---

> ### Author Response · Authors · 2025-08-05
>
> We thank the reviewer for the prompt reply and the consideration of our work. We are sorry to hear that addressing the majority of concerns did not alter the score for our submission.
> As a quick reply to the last questions:
>
> - We appreciate the importance of highlighting additional baselines. As discussed, FNO can be considered an independent baseline similarly to the FEM ground-truth, i.e. as a completely independent, data-informed method. We will consider including its outcome in the revised manuscript.
>
> - For the additional metrics tables, the _nan_ values simply refer to non-applicable methods, see also Table 1 in the original submission for comparison. We will follow the same notation of the original submission and report them as "-" in the revision, thanks for highlighting it.

---

### Note · Authors · 2025-08-14

We thank the Reviewers for a constructive review process: we believe our NSN paper was in general well understood.
Our revision focuses on manuscript clarity, especially regarding baseline comparisons and efficiency, with concrete additions expanding on the scope. We have primarily:
- Drafted an extended discussion of non linear cases where our method can be applied
- Included upon request of one reviewer additional performance metrics
- Elicited with numerical data the favourable comparison of our method's computational cost over its direct competitors.

In particular we highlight:

**Clarity & comparability**
Section 4 is now self contained. Tables mark best and second-best and render non-applicable metrics as “–”. Beyond RMSE, we report relative L2, pressure-recovery error, long-horizon max deviation and total-variation smoothness when applicable.

**Efficiency & training dynamics**
Appendix A.4 includes parameter counts and training curves (loss + validation error) showing faster, smoother convergence than PINNs, reflecting boundary-only supervision and a smaller search space. We also provide exemplar wall-times under identical code/seed/hardware to better highlight the faster training enabled by NSNs.

**Baselines & positioning**
We expand on hard-constraint PINNs and clarify that NSN embeds the solution manifold architecturally rather than enforcing constraints via optimization or domain warping. We also position FNO/operator learning as data-trained surrogates on fixed grids, complementary to NSN’s zero-shot PDE solves.

**Scope & limitations**
All reported experiments use analytical bases; “learnable bases” are explicitly future work. Appendix B.5 consolidates a principled roadmap to extend our ideas to (additional) nonlinear cases. Claims are kept aligned with current evidence.

**Practical take-away**
Across benchmarks, NSN delivers competitive (often best/second-best) accuracy with lower variance, while exhibiting stabler optimization, reduced hyperparameter tuning, and a smaller memory footprint.

Given the _large dispersion in scores_, we concentrated on comparability and reproducibility so that any remaining differences reflect interpretation rather than missing information. Most reviewers commented positively on our efforts, declaring that their concerns had been addressed.

All code, configs, and added materials are in the supplement for full reproducibility.
We thank the ACs for considering carefully the review process of our paper.

---

### Decision · Program_Chairs · 2025-09-17

**Decision:**

Reject

**Comment:**

The reviewers are somewhat split on the paper. The reviewers broadly argue that, while the idea of hard‑coding PDE constraints via a superposition of analytical basis functions is novel and the theoretical analysis is solid (some argue that, others are a little bit more conservative arguing that it is basically a textbook generalization), the empirical contribution falls short of NeurIPS standards. The method only excels on a narrow set of linear problems, often delivers second‑best rather than best performance, and fails to demonstrate any clear advantage over existing PINN or operator‑learning baselines on several benchmark tasks (e.g., Laplace‑1, Heat‑1, Navier–Stokes). Moreover, the approach is fundamentally limited to PDEs that admit a superposition principle; extensions to nonlinear equations remain speculative, with no experimental validation beyond a single Burgers example. The paper also lacks comprehensive baseline comparisons (hard‑constrained PINNs, FNO), runtime analyses on comparable hardware, and robustness studies under noisy data—issues repeatedly highlighted by the reviewers. Some of these limitations have been addressed in the rebuttal phase.

So I went back and forth with this paper and even if I take out the most negative review then the still the paper has several issues as mentioned above (the limited empirical impact, restricted applicability, and missing evaluations) that make it hard for me to recommend acceptance. As such I unfortunately have to recommend reject however I encourage the authors to take the feedback into account and submit an improved version to one of the upcoming venues.